

# Atmospheric highs drive asymmetric sea ice drift during lead opening from Point Barrow

MacKenzie E. Jewell[1], Jennifer K. Hutchings[1], and Cathleen A. Geiger[2,3]

[1]College of Earth, Ocean, and Atmospheric Sciences, Oregon State University, Corvallis, Oregon, USA
[2]Department of Geography and Spatial Sciences, College of Earth, Ocean, and Environment, University of Delaware, Newark, Delaware, USA
[3]Vermont Field Station LLC, Hartford, Vermont, USA

**Correspondence:** MacKenzie E. Jewell (jewellm@oregonstate.edu)

**Abstract.** Throughout winter, sea ice leads open episodically from headlands along the Alaskan coast under the winds of passing weather systems. As leads extend offshore into the Beaufort Sea, they produce ice velocity discontinuities that are challenging to represent in models. Here, we investigate how synoptic wind patterns form large-scale leads originating from Point Barrow, Alaska and influence Pacific Arctic sea ice circulation. We identify 135 leads from January-April 2000-2020 and generate an ensemble of lead opening sequences by averaging atmospheric conditions, ice velocity, and lead position across events. On average, leads open as the winds of migrating high-pressure systems drive differing ice-coast interactions across Point Barrow. Southward winds compress the Beaufort ice pack against the coast east of Point Barrow over several days, slowing sea ice drift. As offshore winds develop in the west, a lead opens and separates the western ice pack from the coast. The eastern ice pack remains in contact with the coast, drifting at half the rate of western ice despite similar wind speeds. As a result, sea ice drifts asymmetrically along the Alaskan coast during these events. Most events occur under north or east-northeast winds, and wind direction relative to the coast controls patterns of opening and ice drift. These findings highlight how coastal boundaries modify the response of the consolidated ice pack to wind forcing in winter. Observed connections between winds, ice drift, and lead opening provide effective test cases for sea ice models aiming to capture realistic ice transport during these recurrent events.

## 1 Introduction

Drift of the consolidated Arctic sea ice cover remains challenging to simulate with dynamic sea ice models, particularly in coastal regions. Sea ice motion is primarily driven by stresses from winds and ocean currents, and modified by internal stresses from ice-ice interactions when sea ice concentrations are high. In winter, sea ice concentrations increase and the ice cover expands, meeting the nearly circumpolar Arctic coastline. During this period of sea ice consolidation, internal stresses from interactions between the ice pack and the coast are readily transferred into the sea ice pack. These internal stresses constrain the dynamic response of the ice cover to forcing from winds and ocean currents, weakening correspondence between winds and ice drift within $O(500\,\mathrm{km})$ of coastlines (Thorndike and Colony, 1982). Internal ice stresses remain elevated until the consolidated ice season ends in late spring as ice concentrations reduce and the ice cover retreats.



In the Pacific Arctic the prevailing pattern of sea ice drift, the Beaufort Gyre (Fig. 1), is an anticyclonic circulation that

moves sea ice and ocean surface waters from the Central Arctic Ocean into the coastal Beaufort, Chukchi, and East Siberian Seas. Together with the Transpolar Drift, a major pathway for Arctic sea ice export into the North Atlantic, the Gyre regulates the basin-scale exchange of sea ice across the Arctic Ocean. Since winds describe the majority of sea ice motion away from coastal boundaries (Thorndike and Colony, 1982), the seasonal circulation structure of the Beaufort Gyre largely mirrors a pattern of anticyclonic surface winds that enclose a semi-permanent relative high in sea level pressure, the Beaufort High

(Walsh, 1978; Serreze and Barrett, 2011). Because the average Beaufort High and Beaufort Gyre patterns are centered in the northern Beaufort Sea during the consolidated ice season (Fig. 2), the eastern and southern flows of the Gyre experience significant coastal constraint. The Beaufort Sea ice pack, for example, is bound to the east and south by the Alaskan and Canadian coastlines during the consolidated season. There, a prevailing westward sea ice drift parallel to the northern coast of Alaska transports thick multiyear ice from the Central Arctic into the Chukchi Sea and beyond (Fig. 1), where the ice is

either entrained into the Transpolar Drift, returned to the Central Arctic, or melted in summer. In recent years, westward sea ice drift in the southern Beaufort Sea has accelerated (Kwok et al., 2013), entraining more multiyear ice into the Beaufort Sea (Babb et al., 2022). Together with increasing rates of ice melt in the Beaufort Sea during summer (Mahoney et al., 2019), these changing ice dynamics have weakened sea ice recirculation toward the Central Arctic, reducing multiyear ice coverage in recent decades (Kwok and Cunningham, 2010; Kwok, 2018).

Rates of sea ice flux across the Beaufort Sea are highest during the consolidated sea ice season (Howell et al., 2016). In order to accurately predict continued changes in Arctic sea ice coverage, sea ice transport during the consolidated season must be accurately represented in models. However, current climate model simulations tend to produce unrealistic ice drift speeds compared to observations during the consolidated season (Rampal et al., 2011; Tandon et al., 2018). Ice drift in the southern Beaufort Sea remains especially poorly represented across a range of model formulations (Kwok et al., 2008). Accurate

representations of sea ice volume transport in the Beaufort Sea remain challenged by multiple sources of variability in the mechanisms that control sea ice drift. The seasonal structure of the Beaufort High varies with the type, track, and persistence of synoptic weather systems that comprise it, driving variability in the synoptic wind patterns that force the sea ice cover. Internal stresses from ice-ice and ice-coast interactions modify ice kinematics, introducing spatiotemporal variability in the dynamic response of the ice to forcing. Consequently, motion of the Beaufort Sea ice pack during the consolidated season is

characterized by intermittent transition between periods of relative quiescence and rapid sea ice drift. These transitions occur on the order of hours in association with large changes in internal stresses (Richter-Menge et al., 2002) during large-scale sea ice fracturing events. Landfast ice (quasistationary sea ice that fastens to the coast or shallow ocean floor) acts as a modified boundary condition on ice motion during the consolidated season, typically paralleling the $20\,\mathrm{m}$ isobath along the Alaskan coast (Mahoney et al., 2007). As winds drive ice-coast interactions along these boundaries, sea ice stresses can exceed the

strength of sea ice, causing the ice cover to fracture and sometimes form large-scale quasilinear openings called leads. Headlands and sharp corners in landfast ice along the Alaskan coast often serve as nucleation points for lead formation. As leads open from these coastal sites and extend offshore into the central ice pack, they drive spatial and temporal discontinuities in sea





ice velocity that cause regional ice motion to appear episodic. These events make the nature of the Beaufort Sea ice velocity field challenging to predict and represent in models.

Recurrent patterns of lead opening from the landfast ice edge and promontories along the Alaskan coast have previously been visually identified and categorized based on their location and geometric characteristics (Eicken et al., 2005; Lewis and Hutchings, 2019). These observational analyses indicated that locations of lead opening along the coast and associated patterns of ice flux throughout the Pacific Arctic may be related to the position and persistence of weather systems translating the region. These studies highlighted a need for further investigation into the connections between weather systems and changes in

sea ice circulation that accompany specific patterns of coastal lead opening during the consolidated season. As sea ice models progress toward realistic representations of spatiotemporally discontinuous sea ice motion associated with coastal lead opening in the consolidated ice cover (e.g. Rheinlænder et al. 2022), observational descriptions of the relationships between winds, ice drift, and lead opening could be used to quantitatively evaluate the performance of model simulations.

    The objectives of this study are (1) to determine the nature of sea ice circulation in the Pacific Arctic as large-scale coastal

leads open in the Beaufort Sea during the consolidated season, and (2) identify the range of weather patterns that initiate these recurrent events. We focus on the formation of leads from Point Barrow, a prominent headland along the Alaskan coast that is a site of especially frequent lead activity (Willmes and Heinemann, 2015). We consider lead formation from 2000 to 2020 between January through April because these months fall within the consolidated ice season. By identifying spatial differences in the ice drift response to winds adjacent to the Alaskan coast, we also aim to explore the role that coastal geometry plays

in constraining ice motion during these events, thereby providing insight into the broader role of ice-coast interactions in ice dynamics throughout winter and early spring.

    This paper is organized as follows. In section 2, we describe the methods used to identify Point Barrow lead opening events and observationally characterize the dynamic conditions associated with the events. In section 3, we present the results of the analysis: the range of synoptic atmospheric conditions associated with lead opening at Point Barrow, atmospheric and sea

ice drift changes during an ensemble average lead opening sequence, and controls on lead orientation and sea ice motion by patterns of wind forcing relative to the coast. In section 4, we discuss connections between wind forcing patterns and coastal boundaries during these events, as well as the challenges in interpreting the impacts of the events due to cross-event variability. Finally, in section 5 we summarize our results and their potential in guiding the development of sea ice models that aim to accurately represent drift of the consolidated sea ice cover during these recurrent coastal fracturing events.



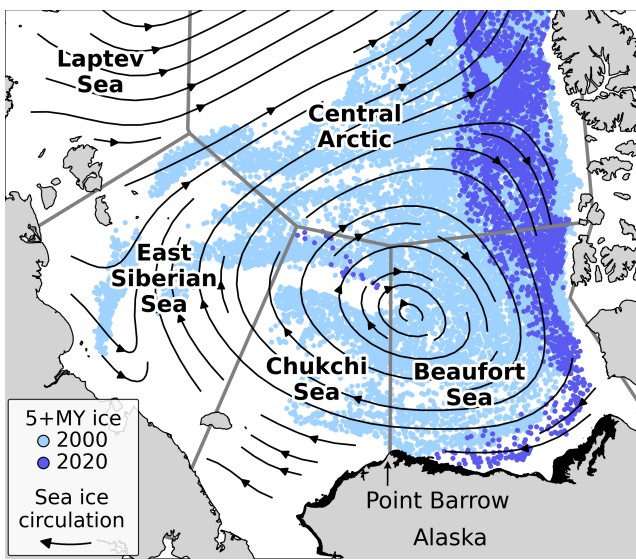

**Figure 1.** Coverage of ice older than five years from the week of January 1 during 2000 and 2020 from Tschudi et al. (2019b). Streamlines of mean January-April 2000-2020 Polar Pathfinder sea ice drift overlain (Tschudi et al., 2019a). Ocean bathymetry 20 m and shallower (GEBCO Bathymetric Compilation Group 2022, 2022) filled in black around the Alaskan coast to demonstrate typical regional landfast ice extent (Mahoney et al., 2007). Grey borders demarcate Arctic Ocean regions, as Fetterer et al. (2010).

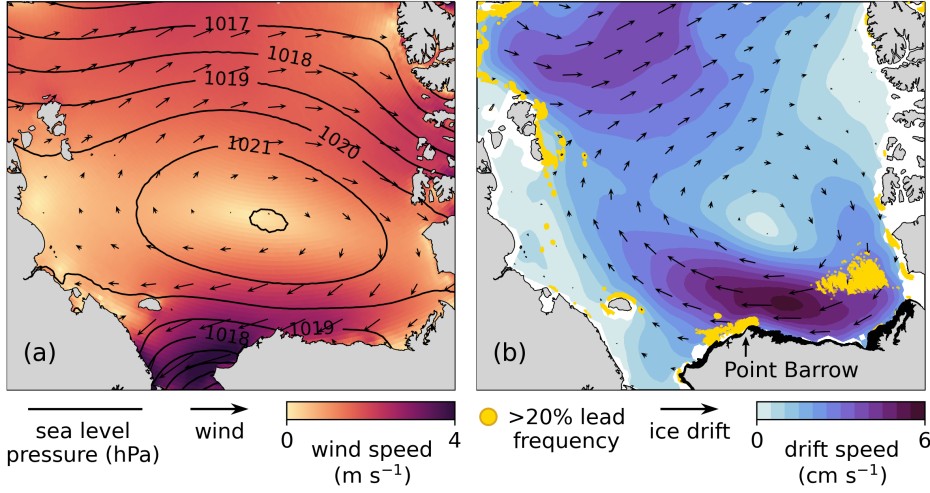

**Figure 2.** Climatological mean atmospheric and sea ice circulation fields (Jan-Apr 2000-2020). (a) ERA5 reanalysis 10-meter winds and mean sea level pressure contours. (b) Polar Pathfinder sea ice drift, overlain with MODIS-detected lead frequencies greater than 20% for November–April 2002-2019 (Reiser et al., 2020). Alaska landfast ice shaded black as in Fig. 1.



## 2 Data and Methods

### 2.1 Satellite imagery data

Leads were identified with Moderate Resolution Imaging Spectroradiometer (MODIS) Level-1B Calibrated Radiances in the longwave infrared band (Band 31: $10.780 - 11.280\ \mu m$). The longwave infrared band is useful for detecting leads due to the thermal contrast between open water or thin ice within a lead and the thicker ice that surrounds it. Under favorable atmospheric conditions and with low viewing angle, a satellite's thermal imaging sensor may detect leads narrower than its resolution (Stone and Key, 1993). With the $1\ \mathrm{km}$ resolution of MODIS Band 31, lead widths as small as $200\ \mathrm{m}$ may be detectable under ideal conditions (Stone and Key, 1993).

MODIS instruments are on board the Terra (EOS AM-1) and Aqua (EOS PM-1) satellites of the NASA Earth Observing System program (MODIS Characterization Support Team (MCST), 2017a, b, c, d). Together, the satellites pass over the Beaufort and Chukchi Seas approximately every 8 hours. We acquired two daily composite images (median times 06Z and 22Z) of the study region beginning February 2000 following Terra's launch. Aqua's launch in spring 2002 provided an additional overpass (median 13Z), increasing the collection to three times daily until the end of the analysis period in 2020.

### 2.2 Atmospheric reanalysis data

We used ERA5 atmospheric reanalysis (Hersbach et al., 2018) from the ECMWF to analyze atmospheric conditions. ERA5 has $0.25°$ spatial resolution and hourly temporal resolution. We used hourly mean sea level pressure (SLP) and 10-meter wind data for the months January through April from 2000 to 2020. Daily averages were calculated from the hourly data. In an intercomparison of six atmospheric reanalyses against independent in situ observations, ERA5 most accurately reproduced observed 10-meter wind speeds over Arctic sea ice in winter and spring (Graham et al., 2019), with a root-mean-square error (RMSE) of $1.4\ \mathrm{m\,s^{-1}}$ from January-March and $1.1\ \mathrm{m\,s^{-1}}$ in April-May.

### 2.3 Sea ice drift data

Sea ice velocity was taken from the Polar Pathfinder Daily 25 km EASE-Grid Sea Ice Motion Vectors data product (Tschudi et al., 2019a). The Polar Pathfinder drift product is generated using an optimal interpolation of remotely sensed sea ice drift data calculated from passive microwave sensors (AMSRE, SMMR, SSMI, and SSMI/S), in situ drift data from International Arctic Buoy Program (IABP) buoys, and drift estimated from NCEP/NCAR Reanalysis 10-meter winds. In an intercomparison of four sea ice drift products, the Polar Pathfinder product was shown to have the highest correlation with buoy drift data and a RMSE of $1\ \mathrm{cm\,s^{-1}}$ in winter (Sumata et al., 2014).

### 2.4 Lead identification and extraction from MODIS imagery

External forcing (atmospheric forces, oceanic forces, and boundary forces from coastlines and bathymetry) and internal forcing (dynamic ice-ice interactions and material heterogeneities) each play a role in the timing and direction of leads forming from



Point Barrow. The degree to which each of these variables contribute varies across lead opening patterns. Since the mechanisms of wind-driven lead opening were the focus of this study, we analyzed lead patterns that formed with a specific set of geometric constraints to isolate controls by winds.

Point Barrow is the main seaward coastal boundary prominence along the north Alaskan coast. Other geographic and bathymetric features are known to place considerable constraint on some patterns of lead opening at Point Barrow. These patterns

include small-scale arches that extend westward from Point Barrow to Hanna Shoal (a shallow region $\sim 150\,\mathrm{km}$ offshore) and flaw leads which open southeastward along the semi-stable landfast ice that parallels the Alaskan coast, typically along the 20 meter isobath (Mahoney et al., 2007). To omit these patterns of opening, leads were only included if they formed cohesive patterns extending northward from the landfast ice near Point Barrow into the offshore pack ice.

Pre-existing deformation features (e.g. leads and ridges) introduce heterogeneities into the sea ice cover which influence

its local mechanical properties. Multiple leads will often form at Point Barrow in rapid succession, either as distinct lead patterns or as a system of leads branching from the same initial fracture as it deforms. This activity modulates the local ice thickness distribution and can influence subsequent openings. Since the ice cover is continually deforming, impacts of such heterogeneities cannot be completely eliminated from this analysis. However, to reduce contributions from previous deformation, we only selected leads that were considered to have formed anew by following a distinct path offshore from

any existing leads at Point Barrow. In the case that a system of branching leads had formed between successive images, the bounding branch of the lead system (e.g. easternmost lead in a system of leads advancing eastward) was collected.

Once the constraints on selecting Point Barrow lead patterns were established, leads were visually identified from three daily MODIS images over the analysis period. In images with low cloud cover, the kinematic sea ice conditions were categorized and the image was visually inspected for lead activity (Appendix A). Though cloud cover is at a minimum during winter and

early spring (Intrieri et al., 2002; Shupe et al., 2011), cloud cover was too extensive in $53\%$ of the analyzed images to determine lead presence at Point Barrow. New lead events were recorded for each low-cloud image in which a lead was present at Point Barrow if the lead satisfied the previously defined geometric constraints and was not present in the preceding image in which the ice cover around Point Barrow was also visible. In total, 135 lead events were identified.

We used an active contour model (van der Walt et al., 2014) to semi-automatically extract lead coordinates from thermal

MODIS imagery for each event (Jewell, 2023). Active contour models use iterative spline fitting to search for the strongest data contours through a set of given coordinates. These models are useful for extraction of lead coordinates from thermal infrared imagery, in which relatively warm leads manifest as contrasting lines against the colder sea ice cover. Supplied with a number of manually selected imagery coordinates along each lead pattern from its connection to landfast ice near Point Barrow to its terminus, the active contour model extracted the remaining coordinates which were recorded with $5\,\mathrm{km}$ geodesic spacing (Fig.

3). The MODIS images from which leads were extracted were mapped with an extent between $68.5 - 78°\mathrm{N}$ and $125 - 165°\mathrm{W}$ (using a polar stereographic projection, central longitude $145°\,\mathrm{W}$). The northernmost or westernmost coordinates of a few lead patterns which extended outside the study region were omitted.



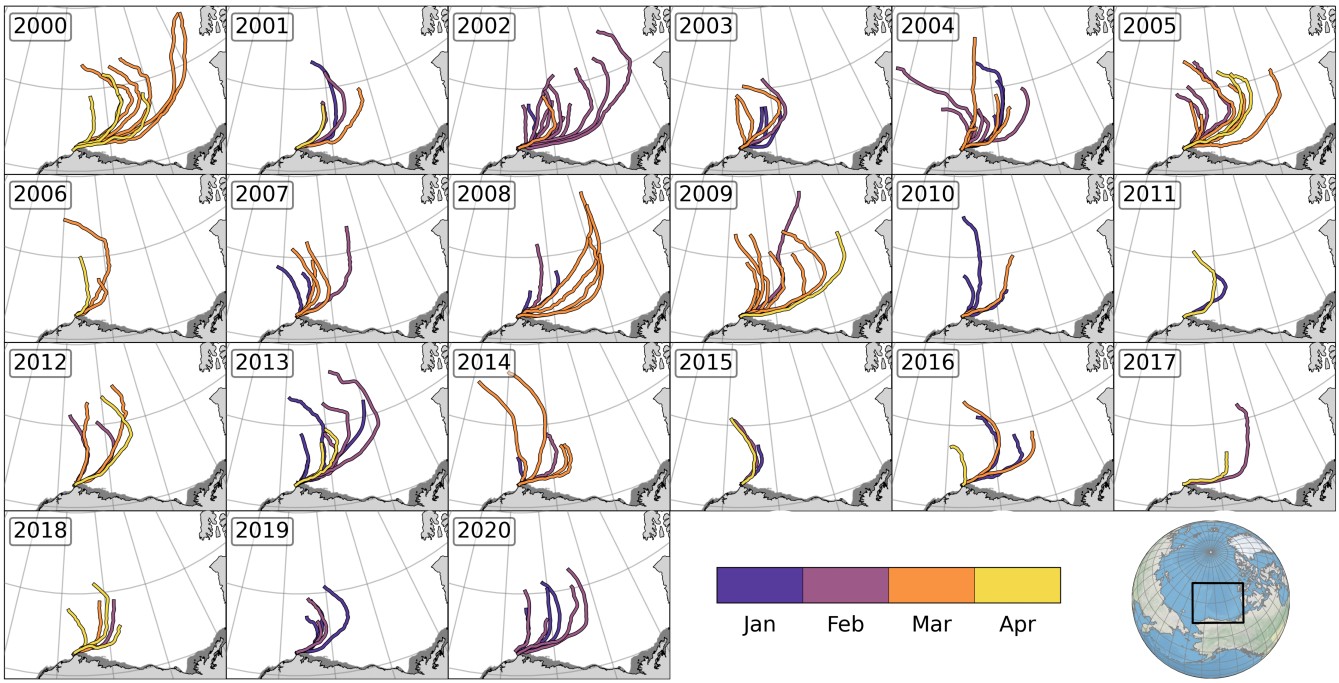

**Figure 3.** Leads identified in each year of the analysis. Lead color indicates the month when the lead formed: January, February, March, or April. Lead widths not shown to scale. Alaska landfast ice shaded grey as in Fig. 1.

## 2.5 Calculation of atmospheric conditions during lead opening

The temporal window and spatial extent over which atmospheric forcing is relevant to lead formation at Point Barrow may
vary across events. To characterize relevant atmospheric conditions, we first considered a region over the Beaufort and Chukchi
Seas between $70.5° − 78°$ latitude and $187° − 220°$ longitude, which we refer to as the BCS region (Fig. 4). The BCS region is
centered zonally around Point Barrow and extends meridionally from where the ice pack contacts the coast near Point Barrow
to the center of the climatological (January-April 2000-2020) mean position of the Beaufort High. SLP and 10-meter winds
were spatially averaged in this region around the time of each lead formation. The time of lead formation cannot be precisely
ascertained from the analyzed imagery for two primary reasons. First, a variable time lag follows each sea ice failure as it
widens sufficiently (at least $200\,\mathrm{m}$) to become visible in thermal MODIS imagery. For instance, a narrow shear failure may
not become visible in satellite imagery until it is widened by winds (Overland et al., 1995). Second, a time lag separates initial
lead detectability and lead identification since leads were identified from three daily images. In this analysis, each lead was
assumed to have formed at some point between the image in which it was identified and the preceding clear sky image. The
average time gap between successive analyzed images is approximately eight hours. To determine the atmospheric conditions
during the approximate time of lead opening, they were averaged over a 9-hour period preceding and including the hour of the
image in which each lead was collected.





## 2.6 Construction of an ensemble lead formation event

We evaluated the typical dynamic conditions throughout the process of lead formation at Point Barrow by constructing an
ensemble average sequence of daily atmospheric conditions, sea ice motion, and lead location across events. Identified events
were aligned in time by the day of lead opening (DLO) and cross-event average atmospheric and sea ice conditions were
calculated from three days before to two days after the DLO. This six-day period was chosen due to similarity in synoptic
atmospheric conditions across individual events within this window (Appendix B1). Many event sequences overlap since
multiple leads will often form in response to the same weather system. To eliminate overlap across event sequences, sequences
were only included in the ensemble if they did not overlap with any of the earlier six-day sequences. This reduced the ensemble
size to 82 distinct lead opening sequences, though the results are qualitatively similar when all 135 events are included in the
ensemble.

For each day of the ensemble event sequence, daily SLP, wind, and sea ice drift anomalies were calculated relative to the
climatological mean conditions (from daily mean conditions January-April 2000-2020). Vector anomalies were calculated by
subtracting climatological vector components from the daily ensemble event vectors. For portions of the analysis in which wind
speeds were directly spatially compared to sea ice drift speeds, wind data were linearly interpolated to the 25 km EASE-grid
on which the Polar Pathfinder sea ice data are gridded.

To determine how the atmospheric conditions and sea ice circulation in the ensemble relate to the average location of lead
opening, the zonal mean lead position was calculated as a function of latitude across the 82 lead patterns from distinct events.
Because leads vary in length and orientation as they extend offshore, the mean and standard deviation in zonal lead position
become increasingly variable with increasing latitude (Appendix B2). The ensemble lead was therefore only calculated up to
the latitude where 70% of the leads extend.

## 3 Results

### 3.1 The atmospheric conditions that precede lead opening

A unique signature is found in the synoptic atmospheric conditions over the BCS region in the hours during lead opening
at Point Barrow. Figure 4 shows the wind and SLP distributions during opening of the 135 identified lead patterns. The
distributions show that Point Barrow lead formation is associated with two primary wind directions (in contrast with the
climatological wind distribution) and higher-than-average SLP over the BCS region, even relative to a high pressure bias in
cloud-free imagery.

The climatological (January-April 2000-2020) distribution of SLP in the BCS region is centered around $1020.6 \pm 10.9\,\mathrm{hPa}$.
Across the 9-hour windows associated with formation of the 135 identified lead patterns, the mean SLP increases to $1027.7 \pm$
$9.9\,\mathrm{hPa}$. Half of this pressure increase from the climatological mean may be attributable to a high-pressure bias introduced
from the low-cloud requirement for lead identification (Appendix C). However, the other half constitutes an increase in the
pressure distribution relative to those of both the climatology and the low-cloud imagery. It also co-occurs with a reduction in





pressure variability relative to climatology and low-cloud imagery. This indicates that lead formation is usually associated with higher-than-average SLP over the BCS region.

Winds usually blow from the east-northeast over the BCS region (Fig. 4b) due to the climatological position of the Beaufort High (Fig. 2). In the climatological distribution, winds originate from the east quadrant the majority (42%) of the time. Northerly (23%) and westerly (21%) winds are next most frequent, and southerly winds (14%) are relatively uncommon.

During lead events, the distribution of wind directions diverges from the climatology. In contrast to the climatological wind distribution, winds during lead opening events are primarily described by two distinct peaks from the east-northeast and from the north directions. Consequently, winds blowing from the north and east become more common during lead formation than in the climatology, preceding 54% and 30% of events, respectively. Southerly (11%) and westerly (5%) winds each become less frequent. Westerly winds feature the largest change in frequency between the climatology and lead events, occurring four times

less frequently during lead opening events than usual. Thus, although westerly winds are relatively common in the region, they rarely initiate lead opening at Point Barrow.

The requirement for low cloud cover does not bias the sampling of wind direction associated with lead opening. This is evidenced by the similar wind distribution between climatology and low-cloud imagery (Appendix C). The frequencies of winds blowing from each cardinal quadrant in cloud-free images are within 5% of the climatological occurrences. Thus, little

of the difference between wind directions when leads are present or not can be attributed to our method of lead detection. In contrast to the large changes in wind direction between climatology and the identified lead events, median wind speed in the BCS region varies little between the climatology ($4.8\,\mathrm{m\,s^{-1}}$), low-cloud images ($4.5\,\mathrm{m\,s^{-1}}$), and lead events ($4.6\,\mathrm{m\,s^{-1}}$).

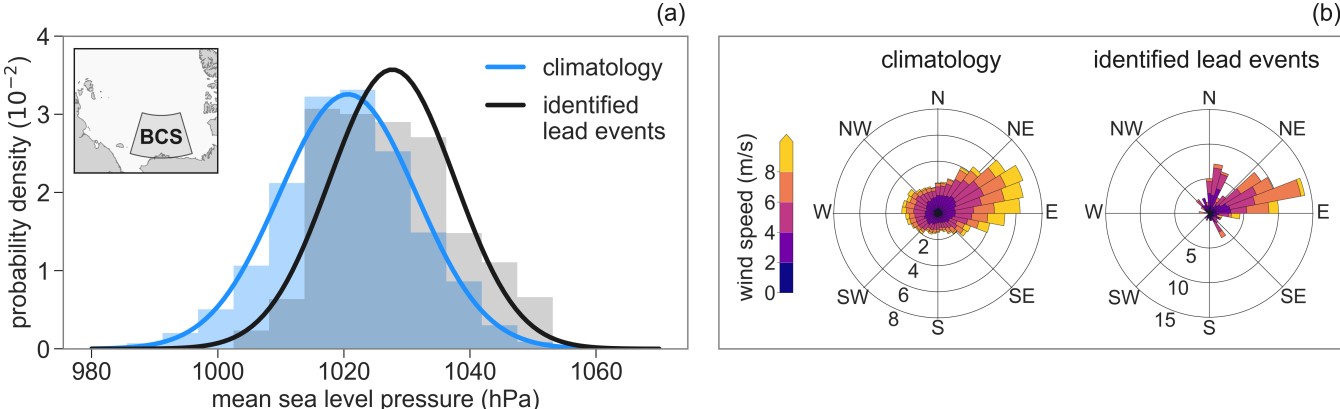

**Figure 4.** (a) Distribution of mean SLP in the BCS region (inset map) across the 9-hour windows preceding identified lead events and across the climatology (all hours January-April 2000-2020). (b) Wind roses showing the distribution of wind speed and direction (from which the wind blows) for the mean wind vector of the BCS region, across the climatology (left) and lead events (right). Radial labels indicate the percentage of wind directions that fall in each sector. Colors indicate the wind speed distribution within each sector.





## 3.2 Typical atmospheric and sea ice circulation during lead opening at Point Barrow

### 3.2.1 Ensemble average lead opening sequence

In the ensemble average event, a high-pressure weather system travels northeastward across the western Arctic, driving lead opening from Point Barrow and producing asymmetric ice drift between the Beaufort and Chukchi Seas. Figure 5 shows the sequence of atmospheric conditions and sea ice drift in the days before, during, and after the DLO in the ensemble lead opening event.

In the days preceding lead opening, a high-pressure atmospheric system travels northeastward from the East Siberian Sea
towards the Beaufort Sea, strengthening winds from west to east along the Alaskan coast. The anticyclonic wind pattern and resultant anticyclonic sea ice circulation resemble the Beaufort High and Beaufort Gyre circulations, respectively, but both patterns are offset westward from their average January-April positions. Over the Beaufort Sea, northerly winds meet the Alaskan coast at a high angle, moving ice southward toward the coast. Restricted by this rigid boundary, the Beaufort ice pack drifts southward slowly at speeds below $4 \mathrm{~cm} \mathrm{~s}^{-1}$. In the Chukchi Sea, northeasterly winds strengthen and rotate clockwise
over time. In response, the ice cover accelerates and begins to drift northwestward around the ice gyre as the winds move ice along and away from adjacent coastlines.

On the DLO, the ensemble average high pressure resides over the northern Chukchi Sea and reaches its maximum pressure. The centers of the wind and ice circulation patterns are collocated with their climatological positions on this day, but the shapes and strengths of the circulation patterns differ from the climatological patterns. Under sustained northerly wind forcing over the
Beaufort Sea and strengthening easterly winds over the Chukchi Sea, the ice pack opens from the landfast ice edge near Point Barrow as a lead extends northeastward into the ice pack between the Beaufort and Chukchi Seas. Ice drift speeds accelerate downwind of the fracture where the ice pack loses contact with the coast, while the upwind drift speed remains weak. As a result, a strong zonal gradient in sea ice drift speed develops in the region between the high-pressure weather center and the Alaskan coast.

In the days following lead opening, winds strengthen and shift westward over the Beaufort ice pack as the high-pressure system continues traveling eastward. The zonal drift speed gradient between the Beaufort and Chukchi Seas reduces as the rapid westward ice drift previously contained downwind of the lead extends eastward toward the Canadian coast of the Beaufort Sea. Eastward progression of accelerated drift speeds is often associated with additional lead patterns opening stepwise eastward from other headlands along the Alaskan coast, a recurrent behavior that occurs as anticyclones transit the region (Eicken et al.,
2005; Lewis and Hutchings, 2019). Before the DLO, some of the westward ice export from the Beaufort Sea was replenished by a southward flow of ice into the northern Beaufort Sea. During the DLO, westward ice export rapidly increases while northern import weakens. Following opening, the ice drift pattern loses a closed circulation structure altogether as motion of the northeastern ice pack stalls and the ice motion regime transitions to an export-dominated flow of ice from the Beaufort Sea.





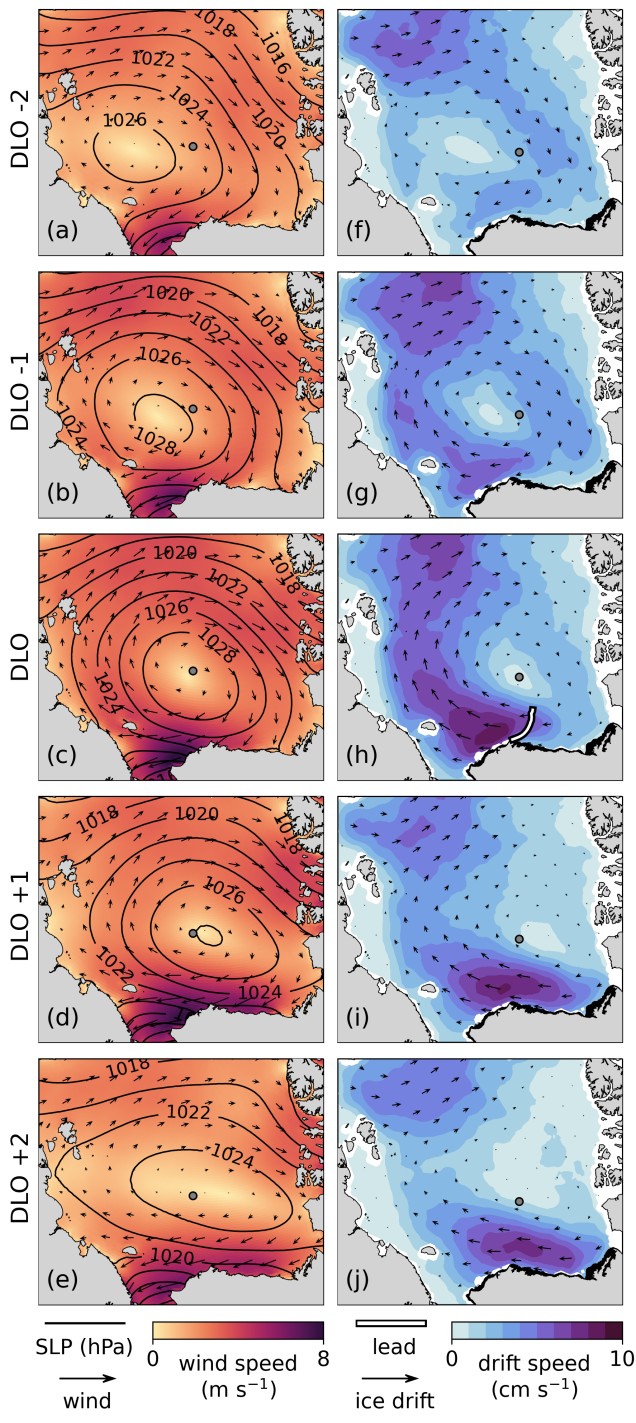

**Figure 5.** Ensemble mean atmospheric conditions (a-e) and sea ice circulation (f-j) over five days of the ensemble lead opening event (white line on h), centered around the DLO. Alaska landfast ice shaded black as in Fig. 1. Mean lead width not shown to scale.





### 3.2.2 Cross-event changes in ice drift across the position of lead opening

To determine whether the spatial differences in sea ice drift from the ensemble event are also reflected across individual events, we calculate the distributions of wind speeds and sea ice drift speeds in two equal-area regions ($2 \times 10^5$ km$^2$) in the Chukchi and Beaufort Seas across event sequences (Fig. 6). The regions are separated by the mean lead position, with the Chukchi (C) region positioned typically downwind of the mean lead and the Beaufort (B) region located typically upwind.

Figure 6 shows the daily median wind and ice drift speeds in either region calculated across the 82 distinct event sequences, aligned in time by the DLO. Results are qualitatively similar when calculated across all 135 events. The climatological daily speed distributions are also calculated in each region for comparison. The event distributions show that asymmetric ice drift in the vicinity of a lead at Point Barrow is seen not only in the ensemble event, but also is reflected across individual events by differences in drift speed across the mean location of lead opening. Further, they demonstrate that wind speed is remarkably similar between the two regions across events, and therefore cannot explain the regional differences in ice motion.

In the climatological conditions, wind speeds in the Chukchi and Beaufort regions are similar, with Chukchi wind speeds on average $9\%$ faster than Beaufort winds speeds. The relative speed difference between the two regions is slightly greater for ice drift, with Chukchi sea ice drifting $20\%$ faster than ice in the Beaufort region on average. The enhanced drift speed difference may reflect the closer proximity of the Beaufort region to adjacent coastal boundaries, resulting in increased ice-coast interactions that restrict ice motion under typical circulation conditions.

Across the lead events, wind speeds in the two regions remain similar as they do in the climatology. From three days before the DLO until the DLO, average wind speeds in the Chukchi region remain within $6-12\%$ of average Beaufort wind speeds. In contrast, the regional ice drift speeds diverge considerably during the event, particularly on the DLO. During the three days before opening, the mean Chukchi drift speed across events remains within $24-36\%$ of the mean Beaufort drift speed. On the DLO, ice in the Chukchi regions drifts at more than double the speed of ice in the Beaufort region, despite similar wind speeds. Zonal asymmetry in ice drift, but not wind speeds, along the Alaskan coast becomes greatly enhanced. Following lead opening, drift speeds in the Beaufort Sea begin to increase while Chukchi drift speeds decrease, reducing the ice drift asymmetry and causing the zonal drift speed difference to fall back towards the climatological value.





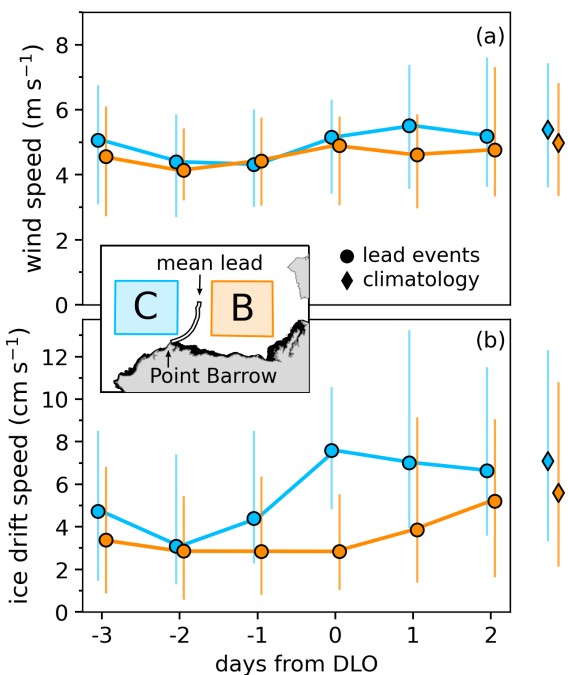

**Figure 6.** Speed distributions of average (a) wind and (b) ice drift in regions C (blue) and B (orange) shown in inset map. Alaska landfast ice shaded black as in Fig. 1. Circles show median across the 6 days of the 82 distinct time-aligned lead event sequences. Diamonds show median across all days January-April 2000-2020 (climatology). Vertical bars show 1st and 3rd speed quartiles.

### 3.2.3 Anomalies in wind-ice relationships during the ensemble event

As demonstrated in Fig. 6, wind speeds do not explain the observed differences in sea ice drift speeds between the Beaufort and Chukchi Seas during Point Barrow lead opening events. Instead, as demonstrated in Fig. 4, wind direction appears to be the primary difference in wind forcing between climatology and the lead events. In Fig. 7, anomalies from climatology during the ensemble average event demonstrate how stronger-than-average zonal SLP gradients tend to shift winds southward during these events. The resultant differences in patterns of wind forcing relative to the Beaufort and Chukchi coasts over time appear to impact the efficiency of the ice drift response to winds adjacent to these coastlines, explaining the zonal ice drift asymmetry associated with lead opening.

Figure 7(f) shows the ratio of the climatological mean ice drift speed ($\overline{U}_i$) to the climatological mean wind speed ($\overline{U}_a$) throughout the Pacific Arctic, $\overline{\alpha} = \overline{U}_i\,\overline{U}_a^{-1}$. Throughout much of the region, the climatological ratio is near $\overline{\alpha} \approx 1.5\%$. Very near to coasts and where winds push the thick multiyear ice cover of the Central Arctic against the Canadian Archipelago, $\overline{\alpha}$ is smaller as the ice is less responsiveness to wind forcing. Under the predominantly meridional SLP gradient north of Point Barrow, $\overline{\alpha}$ is slightly enhanced compared to other regions as ice drifts rapidly westward parallel to the coast. Under this prevailing forcing pattern, sea ice in the Beaufort and Chukchi Seas appear to exhibit similar responsiveness to winds.





During the ensemble event, the spatial pattern of the ice-wind speed ratio deviates from the climatological pattern. The speed ratio for the event is calculated as $\alpha = \overline{u}_i\,\overline{u}_a^{-1}$, where $\overline{u}_i$ is the ensemble mean drift speed and $\overline{u}_a$ is the ensemble mean wind speed. The anomaly in the ice-wind speed ratio from climatology is calculated as $\alpha' = \alpha - \overline{\alpha}$ and shown in Fig.

7(a-e) for $DLO - 3$ through $DLO + 1$ ($DLO + 2$ is not shown as it is similar to the pattern on $DLO + 1$). Anomalies in SLP (SLP') and ice drift vector anomalies are also overlain. To reduce clutter, wind vector anomalies are not included in the figure. Their directions can be visually estimated from the SLP' contours, as they are typically rotated slightly counterclockwise from geostrophic alignment along SLP' contours.

In the days preceding lead opening ($DLO - 3$ through $DLO - 1$), a SLP' high resides over the East Siberian Sea that is

offset southwest from the mean January-April Beaufort High position. The SLP' gradient is predominantly zonal across the Beaufort and Chukchi Seas, producing southward ice drift anomalies toward the Alaskan coast. The ice drift anomalies align with local SLP' contours throughout most of the region, except in the southern Beaufort Sea where the consolidated ice pack is restricted by the rigid Alaska coastal boundary, and ice drift is deflected eastward along the coast. Ice drift is less responsive to wind forcing throughout most of the Pacific Arctic under this synoptic forcing pattern, particularly in the coastally-bounded

Chukchi and southern Beaufort Seas.

Between $DLO - 3$ and $DLO$, the SLP' high migrates eastward. By the $DLO$, zonal gradients in SLP' still dominate throughout the region. However, because the SLP' high is positioned slightly west of Point Barrow, the pattern of anomalous synoptic forcing begins to meet the Chukchi and Beaufort coasts of Alaska differently. East of the SLP' high, a zonal pressure gradient anomaly persists in the Beaufort Sea and continues to drive southward wind anomalies which force the ice pack against the

coast. An anomalously weak ratio of ice drift to wind speeds remains in the Beaufort Sea. In contrast, a positive anomaly in the ratio of ice drift to wind speeds develops in the Chukchi and East Siberian Seas. In these regions that now are positioned south and west of the SLP' high, meridional and zonal gradients in SLP' drive westward and northward wind anomalies. Under this drift-favorable wind pattern, ice drift in these regions becomes temporarily more responsive to winds than usual. The position of lead opening demarcates the regions of the ice pack experiencing enhanced and weakened responsiveness to wind forcing.

This approximately parallels the SLP' contour that meets Point Barrow, which separates the against-coast forcing over the Beaufort ice pack from the offshore forcing over the Chukchi Sea.

Following lead opening ($DLO + 1$), the SLP' high travels further eastward and begins to weaken. A meridional SLP' gradient expands eastward across the Beaufort Sea, producing westward ice drift anomalies in both the Beaufort and Chukchi Seas. As the directions of forcing in the two seas become more consistent relative to the Alaskan coast, the differences in ice

drift responsiveness to winds between the two regions reduces. The magnitude of the ice-wind ratio anomaly reduces nearly everywhere as the synoptic loading pattern returns toward the climatological norm.





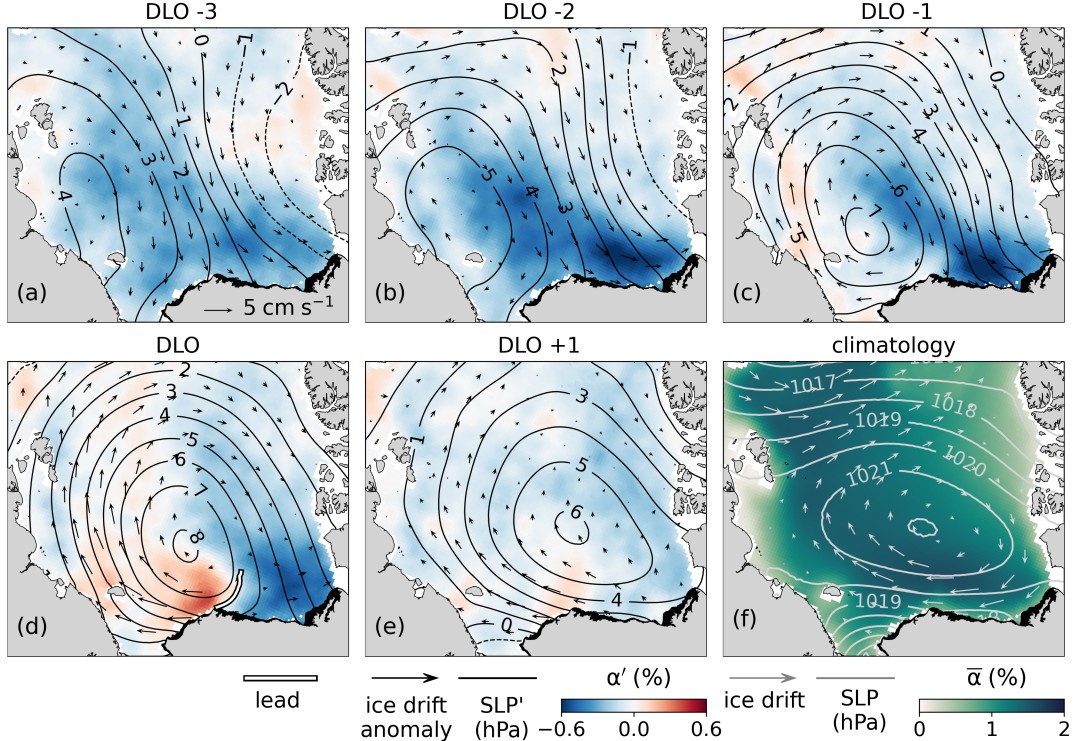

**Figure 7.** Ensemble sequence of anomalies compared to climatology. (a-e) Anomaly in the ratio of ice drift to wind speeds ($\alpha'$) from $\mathrm{DLO}-3$ to $\mathrm{DLO}+1$. Overlain with ice drift vector anomalies and $\mathrm{SLP}'$ contours. Solid (dashed) contours are positive (negative). Mean lead position overlain on DLO (d). (f) Ratio of climatological mean drift speeds to wind speeds ($\overline{\alpha}$). Overlain with climatological mean SLP contours and ice drift vectors. Alaska landfast ice shaded black as in Fig. 1.

### 3.2.4 Contribution of the ensemble lead sequence to climatological ice drift

Leads open from Point Barrow over time scales of hours. However, the identified lead events tend to follow multiple-day sequences of similar synoptic atmospheric conditions. These synoptic patterns drive pronounced deviations in sea ice drift from the mean winter ice circulation. Persistent reductions in Beaufort ice drift speeds relative to wind speeds during these multiple-day lead event sequences result in spatially-varying contributions to the climatological Beaufort Gyre circulation. To demonstrate this, Fig. 8 shows atmospheric conditions and ice circulation averaged over the full six-day ensemble lead sequence. Figure 8(a) shows the six-day average SLP and winds overlain on the ice-to-wind speed ratio anomaly ($\alpha'$) calculated from the average speeds of the six-day sequence. Figure 8(b) shows the six-day average ice circulation pattern. Underlain is the projection of the six-day average ice drift vector anomalies onto the climatological drift vectors. This quantifies how much the ice drift over this period contributes to the climatological ice circulation. Positive (negative) values show where the anomalies are aligned along (against) the typical flow direction, corresponding to a strengthening (weakening) of the typical flow.





On average over the ensemble sequence, ice drifts slightly slower relative to wind speeds than usual ($-0.2\% < \alpha' < 0$) throughout most of the Pacific Arctic (Fig. 8a). Downwind of where the leads form, in the Chukchi Sea, the ice-to-wind speed

ratio shows no appreciable anomaly over this six-day period. Thus, enhancement in ice drifts speeds that occurs on the day of opening (Fig. 7d) is offset by weaker ice drift preceding opening, and the signal of anomalous ice-wind connections is lost over this period. In the Beaufort Sea, however, a clear ice-to-wind speed ratio anomaly does remain over the six-day sequence due to the persistent onshore wind forcing over this period. Winds move ice against the Alaskan coastline east of Point Barrow preceding and during opening, reducing the ice-to-wind speed ratio by more than $0.3\%$ over the full sequence.

As a result of these varying wind-driven ice-coast interactions, the ensemble lead opening sequence's contribution to the climatological Beaufort Gyre circulation varies regionally (Fig. 8b). The change in ice-wind speed ratios across the transition in coastline orientation at Point Barrow produces one of the most striking features of the ensemble event sequence: zonal asymmetry in Gyre strength along the Alaskan coast. The western flow of the Beaufort Gyre is strengthened by $1 - 2\,\mathrm{cm\,s^{-1}}$ on average throughout the ensemble, strengthening ice advection across the Chukchi, East Siberian, and Laptev Seas. Strength-

ening is greatest along the Chukchi coast of Alaska, where the climatological drift is enhanced by over $2\,\mathrm{cm\,s^{-1}}$. In the Beaufort Sea, the Gyre circulation is weakened by $\sim 1\,\mathrm{cm\,s^{-1}}$ during the ensemble sequence. Given the differences between these regions in ice-to-wind speed ratios, the contrasting contributions to the climatological drift appear to arise from differing mechanisms. Weak $\alpha'$ in the western seas indicates that strengthening of the climatological drift does not result from changed responsiveness of the ice to local winds over the ensemble sequence. Rather, the strengthened ice circulation in these regions

results from a typical ice drift response to a strengthening of the climatological wind pattern (not shown). In the Beaufort Sea, $\alpha'$ is strongly negative under sustained onshore wind anomalies. This suggests that negative contribution to climatological ice circulation east of Point Barrow results from weakened ice responsiveness to winds rather than from anomalously weak wind speeds.

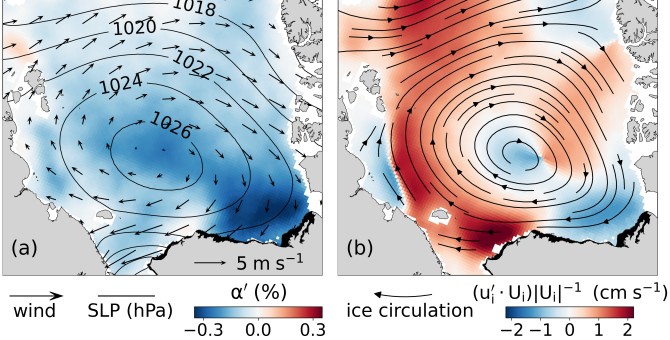

**Figure 8.** (a) Anomaly in the ice-to-wind speed ratio ($\alpha'$) averaged over the six-day ensemble sequence. Average SLP contours and wind vectors overlain. (b) Projection of the average six-day ensemble ice drift anomaly onto the climatological ice drift, overlain with six-day average ice circulation pattern. Alaska landfast ice shaded black as in Fig. 1.



### 3.3 Controls on ice drift and location of opening by wind direction

The ensemble lead opening sequence shows how the migration of a high-pressure weather system on average initiates lead opening at Point Barrow. Across individual events, the positions and structures of high-pressure systems vary, including on the days when leads open. These variable SLP patterns produce a range of wind forcing conditions over the BCS. Each wind pattern drives lead formation at Point Barrow, but does so by initiating different patterns of regional ice drift and orientations of lead extension into the Beaufort ice pack.

To explore the range of wind forcing conditions that can initiate Point Barrow lead opening and their impacts on ice drift and lead formation, we separate events into four types based on the direction of wind forcing over the BCS during the 9-hour window in which leads were assumed to have formed (see the wind distribution in Fig. 4). Each type is defined by a cardinal direction quadrant: those that occur under northerly winds (originating from NW through NE), easterly winds (NE through SE), southerly winds (SE through SW), and westerly winds (SW through NW). For each of these categories, an ensemble 355 average lead event is calculated on the DLO by averaging the daily atmospheric conditions, ice drift, and location of lead opening across individual events. The ensemble average lead events are shown in Fig. 9.

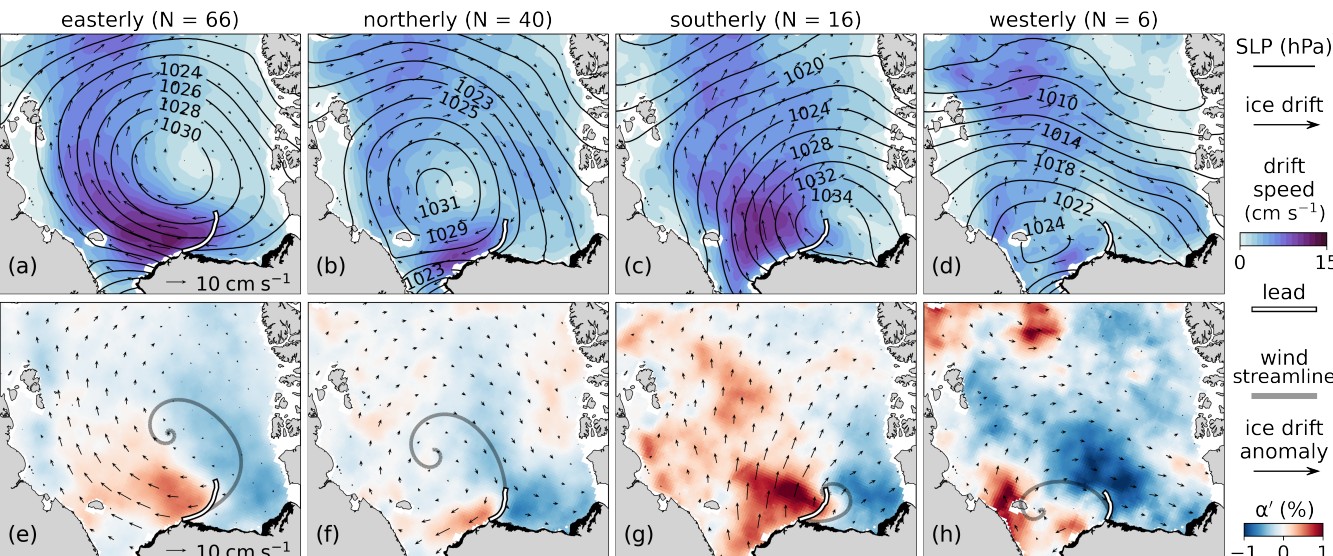

**Figure 9.** Lead ensembles sorted by BCS wind direction: easterly (a,e), northerly (b,f), southerly (c,g), and westerly (d,h). Top row (a-d) shows daily ice drift overlain with SLP contours for each ensemble. Bottom row (e-h) shows anomalies in ice-to-wind speed ratios ($\alpha'$), overlain with ice drift vector anomalies and the wind streamline intersecting Point Barrow. Mean lead position overlain on both rows for each ensemble. Alaska landfast ice shaded black as in Fig. 1.





### 3.3.1   Easterly winds

Lead opening at Point Barrow is most often driven by easterly winds over the BCS region. Easterly winds precede 71 events, corresponding to the largest peak in the Fig. 4 wind distribution. The ensemble average event is calculated across 66 distinct

event days (since some events occur on the same day) and is shown in Fig. 9(a). On average, this event type is associated with a high-pressure system located $800\,\mathrm{km}$ north of Point Barrow. The position and structure of this high determines the location of the wind streamline that intersects Point Barrow (overlain on Fig. 9e), which separates two regimes of forcing over the ice based on the wind direction relative to Alaska's Beaufort and Chukchi coastlines.

On the eastern side of the Point Barrow streamline, winds with an onshore component compress ice in the eastern Beaufort

Sea against the Alaskan coast. As a result, northern ice import into the Beaufort Sea nearly stalls (drift speed $< 2\,\mathrm{cm\,s^{-1}}$) and ice to the south drift slowly westward. On the western side of the streamline, winds direct the ice pack along and away from the Chukchi coast. Unrestrained by coastal boundaries, ice in the western Beaufort and Chukchi Seas drifts rapidly ($> 10\,\mathrm{cm\,s^{-1}}$). This produces a large ice drift speed gradient across the Beaufort and Chukchi Seas.

Near the coast, a lead opens along the wind streamline intersecting Point Barrow. As the lead extends offshore, it turns

against the local wind direction and curves toward the center of the high-pressure system in a concave west arched pattern. Downwind of the lead, ice drifts faster than average relative to local wind speeds (Fig. 9e), increasing transport of ice into the Transpolar Drift. Upwind of the lead, where the ice pack remains restrained by the coast, ice drifts anomalously slowly relative to winds compared to climatological conditions.

### 3.3.2   Northerly winds

Northerly winds are the second most common driver of identified events. These winds precede 42 events and capture the second largest wind direction peak in Fig. 4. The ensemble average conditions are calculated across 40 distinct event days, shown in Fig. 9(b). On average, this event type is associated with a high-pressure system located $750\,\mathrm{km}$ northwest of Point Barrow. The wind streamline intersecting Point Barrow approaches the coast from the north, further west than in the easterly wind ensemble. Consequently, a greater portion of the Beaufort Sea experiences onshore wind forcing relative to that in the easterly ensemble

as northerly winds dominate throughout the Beaufort and Chukchi Seas. Given the proximity of the high to adjacent coastlines in the Chukchi and East Siberian Seas, a smaller region west of the streamline experiences the alongshore-wind forcing that is conducive to ice drift.

The average northerly lead pattern forms in alignment with the wind streamline intersecting Point Barrow along its entire calculated extent. The lead demarcates rapid alongshore ice drift (reaching $10\,\mathrm{cm\,s^{-1}}$) in the Chukchi Sea where the ice-

to-wind speed ratio is enhanced compared to climatology, and weak drift in the Beaufort Sea (below $3\,\mathrm{cm\,s^{-1}}$) where the ice-to-wind speed ratio is reduced. In contrast to the easterly event type, the northerly event features both westward Beaufort Sea ice export in the south and import at the sea's northern boundary. The regional pattern of ice drift is characterized by a closed circulation offset westward from the climatological pattern.





### 3.3.3 Southerly winds

Southerly winds less frequently initiate lead opening at Point Barrow, preceding 16 of the identified events. On average, they are associated with a high-pressure system located $400\,\mathrm{km}$ northeast of Point Barrow (Fig. 9(c)). This is much further east and nearer to shore than the highs in the northerly and easterly cases. The wind streamline meeting Point Barrow approaches the coast from a low angle and over a short extent. Offshore winds dominate across much of the western Arctic, driving anomalously strong ice drift relative to wind speeds downwind of the lead. Directly west of the lead, the ice-to-wind speed ratio is enhanced by more than $1\%$ compared to the climatological ratio. In contrast, drift speeds remain low in the eastern Beaufort Sea (east of the Point Barrow streamline) as the ice-to-wind speed ratio is anomalously low.

The average lead opens at a low angle from the coast and terminates at the center of the average high-pressure system. This forms an arch-shaped lead pattern that opens perpendicular to the local direction of ice drift along most of its extent. Unlike the northerly and easterly cases which feature complete or partial alignment of leads with local winds, the southerly lead does not align with the Point Barrow wind streamline except at its connection to the coast. Rather, it opens west of the streamline where winds move ice offshore from the Beaufort Coast into the Chukchi Sea.

### 3.3.4 Westerly winds

Westerly winds are exceptionally rare during lead opening, preceding just six events; fewer than $5\%$ of identified events despite westerly winds having a climatological frequency of $21\%$. A westerly wind ensemble is included for completeness, but due to the small number of westerly events we caution that the westerly ensemble is less representative of the conditions associated with lead opening at Point Barrow than the other wind ensembles with larger sample sizes. On average, westerly-driven events are associated with a high pressure system centered $600\,\mathrm{km}$ west of Point Barrow (Fig. 9d). Weak winds diverge across the streamline that intersect Points Barrow, blowing primarily along the Chukchi and Beaufort coasts with slight onshore components. A small area in the Chukchi Sea experiences an increased ice-to-wind speed ratio compared to climatology, while nearly all of the Beaufort Sea experiences large reductions in drift speeds relative to winds. The average lead opens at a high angle to the coasts in the vicinity of the Point Barrow wind streamline where a weak zonal drift speed gradient occurs. Ice drift is relatively weak throughout the Beaufort and Chukchi Seas, remaining below $5\,\mathrm{cm\,s^{-1}}$ in most locations.

## 4 Discussion

Here we discuss implications of the observed processes during Point Barrow lead opening events. First, we discuss the sensitivity of ice drift to local wind forcing during these events, and how this may challenge predictions of ice dynamics from atmospheric metrics such as climate indices or atmospheric system positions. Next, we consider differences in ice-wind relationships across Point Barrow during lead opening and how this may reflect influences from wind patterns and coastline geometry on the development of internal ice stresses. Finally, we discuss variability across events and how these events contribute on average to the climatological Beaufort Gyre circulation.



## 4.1 Sensitivity of ice drift to synoptic wind forcing patterns


Variability in Arctic sea ice and ocean circulation is often related to atmospheric forcing using metrics such as the position of the Beaufort High (Regan et al., 2019; Mallett et al., 2021) or climate indices like the Arctic Oscillation (Rigor et al., 2002; Williams et al., 2016; Park et al., 2018). Composite analyses of atmospheric and sea ice circulation have shown that in positive phases of the Arctic Oscillation, changes to the structure of the Beaufort Gyre circulation drive anomalous ice thinning in the

Pacific Arctic (Rigor et al., 2002; Park et al., 2018). However, when assessed in individual years, these atmospheric metrics are only sometimes successful at describing dynamic ice conditions (Williams et al., 2016; Park et al., 2018). In this analysis, we demonstrate the sensitivity of sea ice motion to the structure of synoptic scale wind patterns on time scales of days to a week. By doing so, we offer insight into mechanisms modulating ice drift that challenge descriptions of sea ice circulation variability from metrics such as climate indices or atmospheric system positions.

First, this analysis shows that a specific alignment of a synoptic forcing pattern is not always required to initiate recurrent dynamic ice behaviors. During identified Point Barrow lead opening events, winds over the BCS region primarily originate from the north or east-northeast directions (Fig. 4). The centers of the high-pressure weather systems that drive these winds are broadly distributed over the Arctic Ocean (Fig. B2). Thus, lead opening at Point Barrow can be driven by anticyclones positioned at a range of locations so long as they produce the appropriate wind directions over the BCS region.

Second, even when a synoptic center aligns with a known center of action (e.g. a climatological high position), it can still yield pronounced differences in ice motion from the typical conditions under the known center. In the ensemble lead opening sequence, the center of the average wind circulation pattern on the DLO is aligned with its climatological position (the center of the January-April mean Beaufort High). However, the ensemble atmospheric pattern exhibits a stronger zonal pressure gradient than that of the climatological pressure field, resulting in wind direction anomalies over spatial scales $O(500\ \mathrm{km})$. The

resultant wind pattern drives an anticyclonic sea ice circulation that is aligned with its climatological position (the January-April Beaufort Gyre center). However, the smoothly-varying velocity field of the climatological Gyre's southern flow is disrupted by large-scale fracturing of the ice cover between the Beaufort and Chukchi Seas. A strong zonal asymmetry in sea ice drift speeds develops between the two region that is absent from the climatological flow, resulting in anomalous ice flux patterns throughout the Pacific Arctic.

These observations indicate that patterns of sea ice circulation are highly sensitive to the structures of synoptic atmospheric forcing patterns relative to coastlines. Similar conclusions were reached from a modeling study (Rheinlænder et al., 2022) in which a dynamic sea ice model was used to simulate a large-scale Alaska coastal sea ice fracturing event in the Beaufort Sea (corresponding to Point Barrow leads identified in this analysis on February 20-21, 2013). When forced with different realizations of the same weather system from multiple atmospheric reanalysis products, relatively small differences in atmospheric

forcing produced large differences in both the timing and location of coastal lead opening, as well as the structure of the associated ice velocity fields. The modeled and observed sensitivity of southern Beaufort Gyre ice motion to forcing illustrates why certain atmospheric metrics sometimes fail to describe dynamic ice conditions. Depending on the dynamic ice processes under consideration, the relevant forcing patterns may not be captured by climate indices or Beaufort High position.



## 4.2 The role of coastal interactions

As the ice pack consolidates against the coast in winter, stresses from ice-coast interactions are transmitted offshore and underlying stress levels increase throughout the consolidated ice pack (Richter-Menge et al., 2002). Across a range of sea ice model formulations, sea ice drift during the consolidated season is especially poorly represented in the Beaufort Gyre's southern flow along the Alaskan coast (Kwok et al., 2008). Errors in modeled coastal drift likely arise in part from inaccurate representations of sea ice stress development and transmission from ice-coast interactions. Improvements to the representation

of modeled ice interactions require an understanding of the large-scale stress fields that constrain ice motion in coastal regions. Efforts to understand the nature of the internal sea ice stress field currently rely on sparse in situ stress measurements (Richter-Menge et al., 2002) or interpretations of modeled stress fields (Steele et al., 1997) which rely on ad hoc constitutive relationships between stress and ice kinematics.

Internal ice stresses cannot be directly resolved from the observational data employed in this analysis. However, coupled

with interpretations from previous in situ stress measurements in the Beaufort Sea, the findings in this analysis provide insight into the nature of internal stress states from wind-driven ice-coast interactions. Specifically, this work illustrates the spatial extent over which the Alaska coastal boundary modulates the dynamic response of the consolidated ice cover to anticyclonic wind forcing on timescales of days to a week, and how this coastal drift modulation depends on the structure of wind patterns relative to the coast.

Linear relationships between winds and ice drift successfully describe ice motion in the Central Arctic and in summer when ice concentrations and internal stresses are low. As internal stresses increase during winter, however, the ratio of ice drift to wind speeds typically decreases and correlation between winds and ice drift weakens near coastal boundaries (Thorndike and Colony, 1982). Reduced ice-to-wind speed ratios may therefore indicate the presence of increased ice stresses that reduce the efficiency of wind-to-ice momentum transfer. In the ensemble average lead opening sequence, ice-to-wind speed ratios differ

in time and space depending on the pattern of synoptic wind forcing. Point Barrow marks an abrupt transition between the southwest-northeast orientation of Alaska's Chukchi coast and the predominantly east-west orientation of its Beaufort coast. Contrasting wind-drift relationships in the ice packs adjacent to the coastal boundaries on either side of Point Barrow suggest that Alaska's coastal geometry plays a key role in controlling the dynamic response of the ice pack to forcing through the development of spatiotemporally-varying ice stresses.

Preceding and during lead opening at Point Barrow, southward wind anomalies compress the Beaufort ice pack against the Alaskan coast, resulting in anomalously low ice-to-wind speed ratios $O(500 \, \mathrm{km})$ offshore. The region of anomalously low ice-to-wind speed ratios shown in Fig. 7 may reflect the spatial extent of increased Beaufort Sea ice stresses under the ensemble wind forcing pattern compared to typical conditions. Large ice stresses reflecting compression of the ice pack against the coast have in fact been detected from in situ measurements more than $500 \, \mathrm{km}$ from shore during previous onshore wind events

(Richter-Menge et al., 2002). As the high migrates eastward, wind anomalies shift northwestward over the Chukchi ice pack, directing ice motion away from the Chukchi coast. As a lead opens between the Beaufort and Chukchi Seas, it bounds positive anomalies in the local ice-to-wind speed ratio where the ice pack pulls away from the coast. This likely reflects where internal





ice stresses are temporarily reduced, in agreement with previous in situ measurements showing stress reduces to near zero in the consolidated ice cover when winds direct ice away from local coastal boundaries (Richter-Menge et al., 2002). Following

lead opening, the relationship between wind and ice drift speeds fall back to climatological values as wind forcing returns to the climatological average direction.

The northerly, easterly, southerly, and westerly wind ensembles further demonstrate the variable spatial extents of potential ice stress regimes depending on wind forcing. The portions of the ice pack experiencing differing wind-driven coastal interactions are roughly demarcated by the wind streamline intersecting Point Barrow (Fig. 9e-h). To the east, ice motion is limited

under onshore winds. Leads usually open along or west of the Point Barrow wind streamline. In each wind ensemble, the anticyclonic wind forcing pattern results in a western drift enhancement relative to winds and an eastern drift reduction. Because the orientation and extent of the wind streamline intersecting Point Barrow differs across the ensembles, the portions of the Beaufort and Chukchi ice packs experiencing strengthened and weakened drift efficiency, and the magnitude of ice drift, differ across events.

### 4.3 Interpreting impacts of Point Barrow lead opening events

Cumulative winter and spring circulation of the Beaufort Gyre redistributes ice volume throughout the Arctic and influences regional susceptibility of sea ice to melt in summer. In the Beaufort Sea, ice drift patterns regulate multiyear ice export from the Central Arctic (Babb et al., 2022). In recent decades, net sea ice export from the Beaufort Sea between January and April has been significantly negatively correlated with its sea ice area in the following September (Babb et al., 2019). The episodic

drift and deformation events that dominate Beaufort Sea ice motion during the consolidated ice season challenge predictions of dynamic sea ice change over seasonal timescales and longer.

Although leads open from Point Barrow over a number of hours, they occur on average as part of a multiple-day sequence of anomalous ice-wind relationships and sea ice circulation in association with the migration of an atmospheric anticyclone. Comparison of cross-event SLP variability to climatological SLP variability suggests that synoptic conditions are similar

across Point Barrow lead events from three days before to two days after lead opening. On average over this six-day sequence, the pattern of ice circulation appears similar to the climatological Beaufort Gyre. However, the contribution of the average lead sequence circulation to the climatological Gyre is zonally asymmetric across the position of lead opening from Point Barrow. The western flow of the Gyre is enhanced while the southeastern flow in the Beaufort Sea is weakened. Previous analyses support these observations, showing that average winter ice flux is enhanced in the Chukchi Sea and weakened along

the Beaufort Coast of Alaska on days when large-scale lead patterns originating from Point Barrow are present (Lewis and Hutchings, 2019). The findings in this analysis show also that these regional differences in ice motion are related to differences in ratios of ice drift to wind speeds. In the Beaufort Sea, the ice-to-wind speed ratio is weaker than average under sustained southward wind anomalies that increase compression of the ice pack against the coast. In the Chukchi Sea, directly downwind of the average lead opening, ice-to-wind speed ratio is temporarily enhanced as the lead opens and separates the ice pack from

the coast. The prevalence of lead patterns at Point Barrow during the consolidated season may therefore serve as an indicator of seasonal average ice fluxes in the Beaufort and Chukchi Seas.





We identified approximately one distinct event sequence per month in this analysis. These events may therefore describe approximately one-fifth of the total January-April ice circulation. This is a conservative estimate, as nearly 40% of the total identified lead opening sequences overlapped with the 82 distinct sequences that were investigated. In order to support predic-

tions of dynamic sea ice changes over seasonal timescales, the episodic circulation changes associated with these lead events during the consolidated season must be understood and represented accurately in models. The fidelity of modeled ice drift will depend both on the accuracy of synoptic forcing and the representations of stress transmission from resultant ice-coast interactions.

Across events, there is variability in the track, persistence, and structure of anticyclonic weather systems that have been

found to initiate lead opening at Point Barrow. Due to this synoptic variability, the days that leads open can be followed by a range of dynamic ice conditions. Some events are followed by steady ice export for a number of days following the initial opening. Eastward migration of an anticyclone that increases the extent of easterly wind forcing over the Beaufort Sea can open additional leads from other headlands along the Alaskan coast (Jewell and Hutchings, 2023; Lewis and Hutchings, 2019). Such events often result in breakout, in which the entire Beaufort ice pack separates from the landfast ice along the Alaskan

and Canadian coastlines, causing extensive breakup and increasing ice transport along the Alaskan coast. Point Barrow lead opening events often precede breakouts, but are not always followed by such events with sustained and enhanced ice transport. Some events are followed by a rapid return to quiescent conditions. Given the variability across events, the conditions shown in the ensemble are not necessarily predictive of ice circulation changes during individual events, particularly following lead opening. Further, cumulative sea ice transport can be overwritten by changes in sea ice dynamics over a relatively short period

of time (Babb et al., 2019). Thus, care should be taken in assessing sustained impacts on the regional sea ice cover from the circulation changes associated with these fracturing events.



## 5  Summary and Conclusions

Point Barrow exemplifies Alaska's unique coastal geometry and is a site of exceptionally high rates of lead opening in winter. This paper uses observational and reanalysis data to investigate changes in Pacific Arctic sea ice circulation during the for-
mation of large-scale lead patterns originating from Point Barrow, Alaska. Low-cloud MODIS imagery was used to visually identify 135 Point Barrow leads between 2000 and 2020 during the months January-April when the ice pack was consolidated against the coast. ERA5 atmospheric reanalysis and Polar Pathfinder sea ice velocity were used to construct an ensemble average lead opening sequence from 82 distinct event sequences to describe the average conditions associated with Point Barrow lead opening. Differences in lead opening and ice drift across events were related to patterns of synoptic wind forcing against
the Alaskan coast.

On average, the identified Point Barrow lead opening events are associated with an anticyclone traveling eastward over the Pacific Arctic. The anticyclone features an anomalously strong zonal SLP gradient that forces zonally-varying ice-coast interactions along Alaska's Beaufort and Chukchi coasts. Southward wind anomalies drive ice motion against the coast in the Beaufort Sea over several days. Ice drift speeds reduce as onshore wind stress is increasingly balanced by ice interaction
with the coast and itself. As the anticyclone travels eastward, it aligns with its climatological position but features northward wind anomalies west of its center that move ice away from the Chukchi coast. A lead opens from Point Barrow and extends northeastward into the Beaufort Sea, separating the Chukchi ice pack from the coast. Ice rapidly accelerates downwind of the lead, drifting twice as fast as the upwind ice pack in the Beaufort Sea despite equivalent wind speeds. In the days following lead opening, ratios of ice drift speeds to wind speeds fall back toward climatological values as the regional SLP gradient returns
to its meridionally-dominated climatic norm.

Nearly all identified events are associated with anticyclonic winds, which are the prevailing atmospheric conditions in the region during winter and spring. Offsets in anticyclone position and structure from the climatological mean Beaufort High can produce considerable deviations from the prevailing sea ice circulation pattern during these events. The most notable and consistent feature across events is the development of zonal asymmetry in the ice drift field along the Alaskan coast. Because
the ratio of ice drift to wind speeds is enhanced west of the leads and reduced to the east, the pattern of zonal asymmetry depends on the orientation of lead extension into the ice pack. The leads typically form aligned with or west of the wind streamline that intersects Point Barrow. Thus, the pattern of zonal asymmetry is ultimately controlled by the direction of wind forcing relative the Alaskan coast. The closed anticyclones or atmospheric ridges that drive these events can be positioned at a range of locations, and in most cases they produce winds that meet Point Barrow from the north or from the east. These wind
directions occur during 85% of events, equating to an increase of 20% during lead opening events from their climatological frequency. Increased prevalence of northerly and easterly winds, or reduced prevalence of westerly winds, may therefore be an indicator of increased frequency of lead activity at Point Barrow.

Despite the short time scale of lead opening (order of hours), the atmospheric conditions during the identified events exhibit similarities for a number of days preceding and following lead opening. SLP fields show a reduction in cross-event variability
compared to climatological variability over a six-day window from three days before to two days after lead opening. Cross-





event similarity is strongest preceding and during opening. This indicates that lead opening from Point Barrow is usually associated with atmospheric events occurring at the synoptic time scale. The contribution of the average sea ice drift pattern over this six-day sequence to the climatological Beaufort Gyre circulation varies regionally. The western flow of the Gyre becomes enhanced over this period, while the southeastern flow in the Beaufort Sea is weakened. Weakened flow in the

Beaufort Sea occurs in association with a sustained reduction in the ratio of ice drift to wind speeds under onshore winds. If each event sequences lasts approximately six days, the 82 distinct sequences explored here span one fifth of the overall January-April 2000-2020 ice drift. Given their frequency and regionally-differing contributions to the Beaufort Gyre circulation, the ice transport associated with these episodic deformation events must be represented in models in order to describe the seasonal Gyre circulation.

The observational data explored in this paper highlight wind direction and coastal geometry as key controls on Point Barrow lead formation and associated patterns of sea ice motion during the consolidated ice season. Analysis of the connections between winds and sea ice motion suggest that large internal ice stresses are transmitted offshore from coastal boundaries when winds drive ice motion toward the coast, in agreement with previous in situ observations. Alaska's unique coastal geometry influences the pattern of stress development that drives lead opening by constraining ice motion throughout the pack depending

on patterns of wind forcing. When the ice pack loses contact with the coast during lead formation, downwind sea ice drifts rapidly relative to winds, suggesting a reduction in internal ice stresses. Because the findings in this paper are based on remote sensing and reanalysis data, we are unable to directly resolve the internal stress transmission that is responsible for modulating ice motion and initiating fracturing. However, spatial and temporal differences in the relationships between winds and sea ice drift offer visualizations of the potential spatial extent of the coastal boundary's influence on ice motion over time (due to

changing ice stresses) with changing wind forcing conditions. The ice-wind relationships investigated here are specific to the identified deformation events. However, the findings in this analysis offer more general insight into the processes by which coastal boundaries constrain sea ice motion along the southern flow of the Beaufort Gyre when ice is consolidated to the coast. They also demonstrate how the sea ice drift modulation depends on patterns of synoptic forcing and coastal geometry.

The Point Barrow lead opening events identified in this analysis span two decades of varying conditions, including differing

weather pattern structures, ice thicknesses distributions, and ice deformation histories. Due to the complex interplay between these factors across events, the sensitivity of sea ice motion to each of these conditions cannot be directly tested from the observations in this analysis. Future analysis of these observed events involving the use of dynamic sea ice models would be an important step in teasing out the importance of each of these parameters in driving and constraining ice dynamics in the southern flow of the Beaufort Gyre. The connections between wind patterns, ice drift, and lead opening documented in this

analysis provide a range of conditions to test the robustness of dynamic sea ice models aiming to reproduce realistic sea ice drift during these recurrent events.

*Code and data availability.* MODIS Level-1B imagery data were obtained from the NASA LAADS DAAC (https://ladsweb.modaps.eosdis. nasa.gov/, MODIS Characterization Support Team (MCST), 2017a, b, c, d). ERA atmospheric reanalysis SLP and winds were obtained



from the Copernicus Climate Change Service Climate Data Store (https://doi.org/10.24381/cds.adbb2d47, Hersbach et al., 2018). Po-

lar Pathfinder sea ice drift vectors (https://doi.org/10.5067/INAWUWO7QH7B, Tschudi et al., 2019a) and weekly sea ice age (https://doi.org/10.5067/UTAV7490FEPB, Tschudi et al., 2019b) were obtained from the National Snow and Ice Data Center. Arctic bathymetry data were obtained from the British Oceanography Data Centre (https://doi.org/10.5285/e0f0bb80-ab44-2739-e053-6c86abc0289c, GEBCO Bathymetric Compilation Group 2022, 2022). Timing and coordinates of identified lead patterns, as well as which lead events were used in which ensembles, provided in supporting information. Code used for lead extraction (https://doi.org/10.5281/zenodo.7567150, Jewell, 2023)

accessible on GitHub (https://github.com/mackenziejewell/PointBarrowLead_Extraction).

## Appendix A:  Imagery classifications

To identify Point Barrow lead events, each acquired thermal MODIS image was visually analyzed to document the sea ice activity in the region. The three-daily images were sorted into the following categories. (1) Cloudy: clouds obscure the sea ice around Point Barrow and a significant portion of the Beaufort and Chukchi Seas. In this case, sea ice motion and lead activity

could not be analyzed. If few to no clouds were present, the ice cover around Point Barrow was visually inspected and further categorized. (2) Clear: No lead is visible at Point Barrow and there is little visible activity within the ice cover. (3) New lead: a lead originating from Point Barrow is visible and was not present in the preceding image in which the ice cover around Point Barrow was also visible. (4) Continued lead: a previously-formed lead at Point Barrow is visible, either continuing to develop or re-opening along a lead formed during a recent previous event. (5) Breakup: Significant breakup of the ice cover across

the region, sometimes exposing the landfast ice edge along the coast. Usually few clouds are present, but this activity can sometimes be observed through extensive cloud cover. (6) West: An ice arch or breakup activity is present along the Chukchi coast, sometimes extending eastward across Point Barrow. Figure A1 below shows the imagery categorizations over the full analysis period (January-April 2000-2020), including the 135 documented events where new leads formed at Point Barrow.

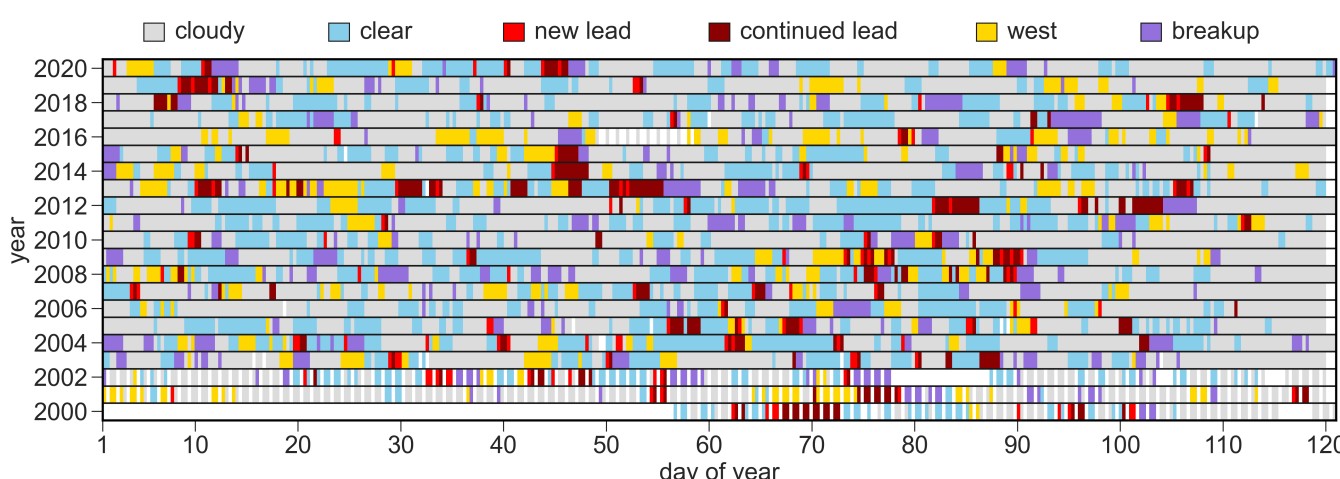

**Figure A1.** Three-daily imagery categorizations over the analysis period. Colors indicate imagery category by year and day of year. White indicates no imagery.





## Appendix B: Ensemble lead sequence calculations

### B1 Determining the ensemble sequence window

The temporal window for the ensemble lead sequence was chosen for consistency in atmospheric conditions across events. To quantitatively evaluate atmospheric consistency in the ensemble, the standard deviation in daily SLP was calculated over the study domain across all distinct individual events in the ensemble from 5 days before to 5 days after the DLO ($\sigma_{\text{event}}$, N = 82). This was compared to the climatological (January-April 2000-2020) standard deviation of daily SLP ($\sigma_{\text{clim}}$, N = 2526 ) at every location to determine which days feature ensemble variability less than the climatological variability. Figure B1 shows the ratio of $\sigma_{\text{event}}$ to $\sigma_{\text{clim}}$ from 5 days before to 5 days after the DLO.

The lowest relative SLP variability of the ensemble sequence occurs on the DLO and on DLO + 1 as $\sigma_{\text{event}}/\sigma_{\text{clim}}$ dips below 0.8 over the central Arctic. The ensemble is therefore most representative of the atmospheric conditions across individual events on the days during lead opening. In most regions $\sigma_{\text{event}}$ increases relative to $\sigma_{\text{clim}}$ with time before and after the DLO, indicating that the quality of the ensemble degrades with time from lead formation. From DLO − 3 to DLO + 2 in the ensemble sequence, the cross-event pressure variability remains lower than the climatological pressure variability over most of the western Arctic Ocean, except over some coastal regions. This period is therefore chosen as the ensemble sequence. Outside this temporal window (on DLO − 5, −4, +3, +4, and +5), $\sigma_{\text{event}}$ exceeds $\sigma_{\text{clim}}$ over portions of the Beaufort, Chukchi, and East Siberian Seas, indicating that the atmospheric conditions are too variable in these regions to be considered in an ensemble.

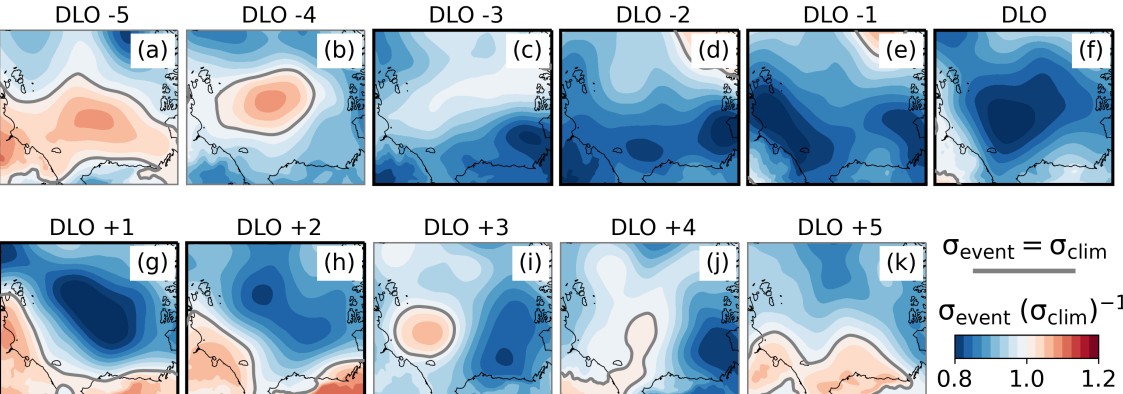

**Figure B1.** Ratio of the standard deviation in daily ensemble event SLP, $\sigma_{\text{event}}$, to the standard deviation in climatological daily SLP (January-April 2000-2020), $\sigma_{\text{clim}}$, for days -5 to 5 from the day of lead opening (DLO) in the ensemble lead sequence. Coastlines are underlain in black. Panels with bold outline (c-h) are chosen for the lead sequence.

### B2 Calculating zonal mean lead positions

For the DLO, the mean and standard deviation in lead longitude were calculated as a function of latitude across identified patterns to quantify the average and zonal spread in the positions of opening. Due to the variable lengths of the lead patterns,



the mean and standard deviation in lead position become increasingly variable as the sample size used in the calculations decreases at higher latitudes (further offshore from their shared position at Point Barrow). To restrict analyses of the ensemble lead position to regions where it is still representative of most events, we only calculate the mean lead in the ensemble up to the latitude to which 70% of the leads used in the ensemble calculation extend. Mean leads are calculated in this way for the full ensemble (Fig. B2a) across leads from the 82 distinct event sequences and for the wind direction ensembles (Fig. B2, b-e) across leads that formed on non-overlapping days.

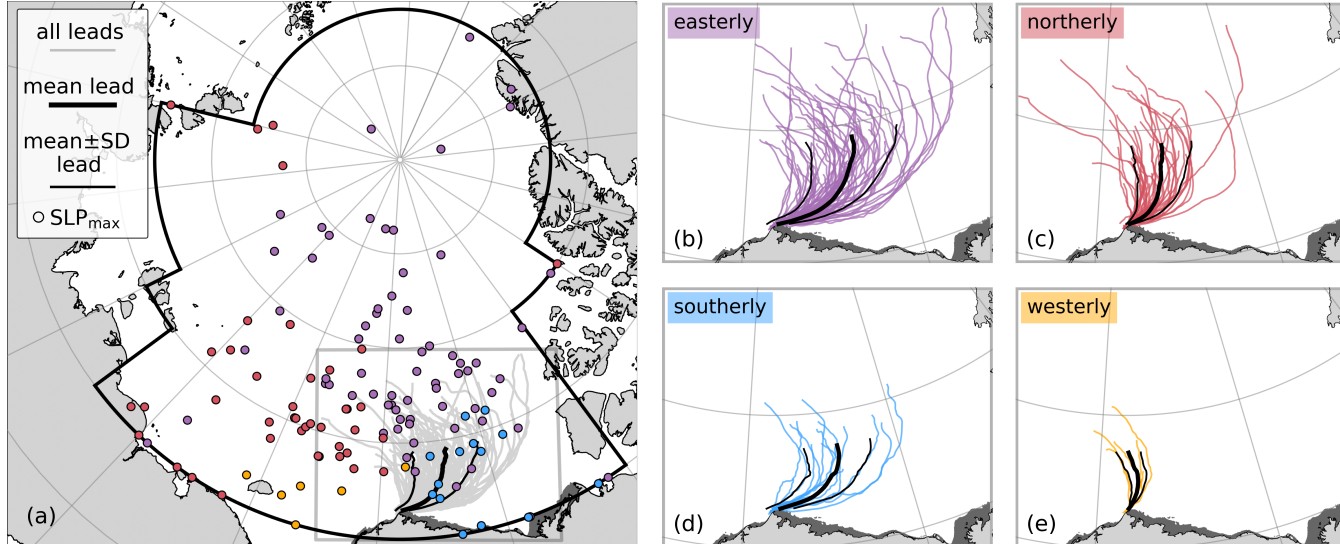

**Figure B2.** Distribution of atmospheric and lead conditions across events, sorted by wind direction. (a) All 135 lead patterns, overlain with mean and mean ± one standard deviation in lead position. Points show positions of $SLP_{max}$ within outlined region from mean SLP fields of 9-hour windows preceding each lead identification. When multiple $SLP_{max}$ are present, the nearest one to Point Barrow is displayed. Points along the region boundary usually correspond to highs or ridges extending over the ocean from land. Dot colors correspond to wind direction sorting in (b-e). Distribution of lead patterns formed under easterly (b), northerly (c), southerly (d), and westerly (e) winds shown as colored lines, overlain with each group's mean lead and mean ± one standard deviation. Alaska landfast ice shaded grey as in Fig. 1.

## Appendix C: Synoptic bias in cloud-free imagery

Identifying leads from cloud-free imagery could bias the synoptic conditions analyzed in the region and therefore those that are determined to drive lead activity. To ensure that the identified differences in synoptic conditions between the lead events and climatological conditions are not artifacts of the requirement for low cloud cover, we compare winds and SLP in the BCS region from the climatology (all hours of the study period January-April 2000-2020) to those during low-cloud imagery and those during the identified lead events. These comparisons are depicted in Fig. C1, which demonstrates that the key differences in synoptic conditions between the climatology and lead events cannot be attributed to the method of lead identification.





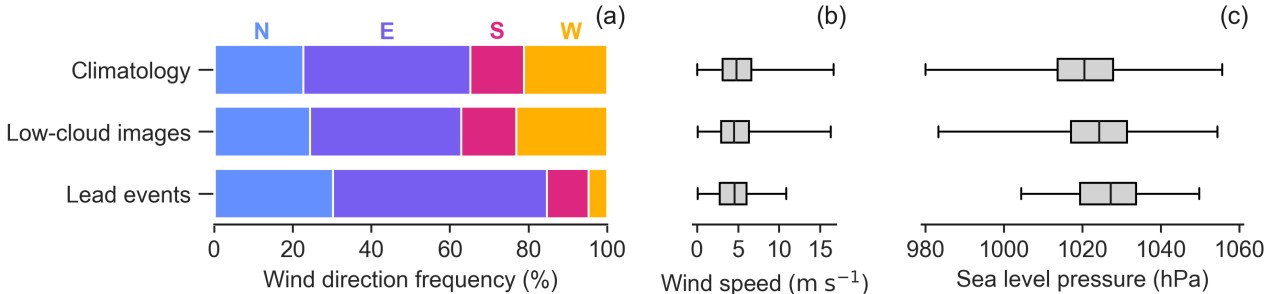

**Figure C1.** Comparison of synoptic conditions between all hours of the study, low-cloud images, and lead events. **(a)** Percentage of time that BCS winds originate from north (N), east (E), south (S), and west (W) directions for each category. Box and whisker plots of **(b)** wind speeds and **(c)** SLP in the BCS region for each category. Whiskers show minimum and maximum values. Boxes show the first quartile, median, and third quartile of the data.

There is little difference between the climatological and low-cloud wind distributions. The frequency of winds blowing from each wind quadrant during low-cloud images remains within $\pm 5\%$ of the climatological values, and the mean wind speed decreases to $4.8 \pm 2.4\,\mathrm{m\,s^{-1}}$ from $5.0 \pm 2.5\,\mathrm{m\,s^{-1}}$ in the climatology. Mean SLP is biased $3.6\,\mathrm{hPa}$ high on average in the low-cloud imagery ($1024.2 \pm 10.6\,\mathrm{hPa}$) compared to the climatology ($1020.6 \pm 10.9\,\mathrm{hPa}$), which is expected given the association between cloudy conditions and decreasing SLP. Between the low-cloud imagery and the lead events, the distribution of wind directions changes substantially. For example, westerly winds during the events are four times less frequent than in the climatology and low-cloud images. In addition, mean SLP increases from that of the low-cloud imagery by $3.5\,\mathrm{hPa}$ (reaching $1027.7 \pm 9.9\,\mathrm{hPa}$). Mean wind speed decreases slightly to $4.5 \pm 2.1\,\mathrm{m\,s^{-1}}$.

The winds during low-cloud imagery are representative of the climatological conditions, wind direction changes considerably during the lead events, and the pressure bias in low-cloud imagery is surpassed during the lead events. We therefore conclude that the synoptic conditions associated with Point Barrow lead events are unique to the lead activity and are not influenced substantially by the method used to identify them.

*Author contributions.* MEJ, JKH, and CAG participated in the conceptualization of the research plan. MEJ developed the methodology, carried out the analysis, and prepared the manuscript. MEJ, JKH, and CAG interpreted and discussed the results, and contributed to the manuscript review and editing.

*Competing interests.* The authors declare that they have no conflict of interest.



*Acknowledgements.* This work was supported by NASA (80NSSC18K1026 and 80NSSC21K1601) and ONR (N000141912604) Grants. We thank Björn Erlingsson for discussions that prompted this investigation.



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
