# Peer review of "Atmospheric highs drive asymmetric sea ice drift during lead opening from Point Barrow"

_The Cryosphere, 2023_

## Author Comment (AC1)

**Response to Referee Comments**

We would like to thank Dr. Willmes for evaluating our manuscript and for providing constructive feedback. With their suggestions the clarity of the manuscript will greatly improve. In this document we will address the comments point by point. Below we show referee comments in orange text and provide our response in light blue text. In dark blue text, we list the changes in the manuscript that address these comments.
* * *
**Referee #1 Comment on tc-2023-9**
Referee comment on "Atmospheric highs drive asymmetric sea ice drift during lead opening from Point Barrow" by MacKenzie E. Jewell et al., The Cryosphere Discuss., https://doi.org/10.5194/tc-2023-9-RC1, 2023

**Summary**
The presented paper aims to evaluate typical dynamic conditions throughout the process of lead formation at Point Barrow. The authors construct an ensemble average lead sequence from MODIS thermal-infrared satellite imagery and derive the associated daily atmospheric conditions and sea ice motion. From this combined data set they find a typical synoptic condition over the Beaufort Sea region during lead opening that is mainly characterized by SLP above average. This pattern appears to cause a strong zonal asymmetry in sea ice drift north of the Alaskan coast, which in combination with coastal interactions, drives the break-up of sea ice with the typical pattern found at Point Barrow. The authors conclude that wind direction and coastal geometry are key controls of lead formation in the Beaufort Sea during wintertime.

**General comments and decision**
The paper represents an interesting study on sea-ice dynamics in the Beaufort Sea during winter and its drivers in the atmosphere. The analysis and the presentation of results are scientifically sound and certainly provide new insight into the causes of sea-ice variability in the Arctic and sea ice - atmosphere as well as sea ice – coastal interactions in general. The study nicely adds up to some other recent publications about what drives the formation of leads in the Arctic and thereby contributes to an improved understanding of the Arctic climate system.
I suggest the paper to be published after mostly minor corrections that I am listing below.

We thank Dr. Willmes for their assessment of the manuscript. Below we will address the minor corrections to the manuscript, and our plan to address these corrections in the revised manuscript.

**Specific comments**

My only major annotation is that the process of the ensemble lead sequence calculation lacks some information to the reader. Although the obtained leads and their patterns are well described in Appendix A1 and B2, it would surely improve the paper if for one exemplary lead sequence the associated satellite images were shown additionally to demonstrate how the observations make up the ensemble. In this context, I am a bit surprised about how exactly the long time series (Fig. A1) was extracted. The authors mention that "each acquired thermal MODIS image was visually analyzed to document the sea ice activity in the region". But that would mean that more than 7000 MODIS composites (3x daily, 120 days, 20 years) were individually screened for the presence of leads? I think that adding the above-mentioned example for some scenes would help clearing this issue. In this context, I also recommend adding a simple graphical demonstration of how the mentioned active contour model (2.4) does extract a lead from the thermal infrared image (raw image and derived lead).

We thank Dr. Willmes for this feedback. We did individually screen more than 7000 MODIS composites for the presence of leads for this analysis. We chose to evaluate this many images (three daily, as opposed to once daily or weekly for example) in an effort to determine the timing of lead openings since the leads can deform or be advected shortly after their formation. We agree that showing an example lead sequence, including the satellite imagery, would make this process more clear. We originally omitted this for considerations of the manuscript length, but as pointed out this would provide clarity for the reader. We therefore incorporate a figure depicting an example sequence in the revised manuscript. This figure (now Figure 3) will show a series of three daily MODIS images during an example lead event, and plots of the associated daily winds and ice drift. We overlay the lead coordinates that were extracted from the image in which the lead was first identified. We think this figure will make our process more clear by showing an example to identify a cohesive lead pattern from the imagery and resulting extracted coordinates.

**Minor comments:**

L 22: "within O (550 km)" What is meant? I guess a technical correction is necessary here.
We will adjust this to state "within approximately 500 km"

Figure 1: A small inset or subfigure with an overview map (whole Arctic) might be useful.
In the revised manuscript, we add an overview map in Figure 1 to show what portion of the Arctic is displayed in the map.

Section 2.4 can be shortened I think. Especially the first two paragraphs seem a bit misplaced.
We thank Dr. Willmes for this comment on the manuscript structure, and will make this section more concise, in part by shifting some of the content of the first two paragraphs into the introduction.

LL 128-130: "However, … Point Barrow". Unclear what is meant here.
This sentence was intended to describe how a set of geometric constraints were used to eliminate leads from the analysis that are understood to form with a strong influence from pre-existing leads. Some leads open in connection to other leads that were already opening at Point Barrow. To eliminate these patterns, whose geometries may be more strongly influenced by the local ice properties than by winds, we included only leads that opened along a distinct path offshore from any existing opening leads at Point Barrow. We will edit the sentence as above to clarify this point, and also restructure the paragraph in which it was written to make this point more clear.

L 156: "200 m". Is that a fixed value determining the minimum width of a lead to become apparent in a MODIS image? Wouldn't that depend on the contrast between lead temperature and surrounding temperature rather than on width only?
We included the phrase "at least 200 m" here to remind readers of the minimum resolvable lead width from 1 km thermal infrared imagery under ideal conditions as described in section 2.1. It is correct that the resolvable lead width would vary depending on the temperature contrast and atmospheric conditions and may sometimes be much larger than this minimum possible value. From this comment we see that including the value here may be more confusing than helpful, so we will remove the phrase "(at least 200 m)" from this section and keep discussions of resolvable leads to section 2.1.

Figure 5: I find it a bit confusing that the lead in the DLO subplot disappears in DLO+1. It might make the reader think that the leads last for one day only.
We thank Dr. Willmes for pointing this out, as it may confuse other readers as well. We initially attempted to address this point by adjusting the figure showing the ensemble sequence by overlaying points where the Reiser et al. (2020) MODIS-derived daily lead data showed a lead present across at least 10% of events in the ensemble on each day of the sequence. However, the figure became very messy and therefore did not show clearly enough the persistence of lead patterns following the DLO. We plan instead to address this point with the figure we will include showing examples of sea ice imagery from an example event, as suggested above, which will show that the leads persist after opening. We will also state this specifically in the ensemble sequence figure caption to clarify this point: "Mean lead (yellow line on h, width not shown to scale) displayed only on DLO although individual openings can persist for longer."

L319: "average speeds" … please add "of sea ice drift"
We will change "calculated from the average speeds of the six-day sequence" to "calculated from the average wind and sea ice drift speeds across the six-day sequence."

L323, L329: These numbers (0.2%, 0.3%) are really small. How does that relate to the effect size? The shown spatial patterns underline that the effect is definitely important, but some discussion about this might help here.

We appreciate this suggestion, and have decided to alter the figures where anomalies in the ice to wind speed ratio are shown in order to more clearly demonstrate the effect size. Where we originally calculated the anomaly in the ice to wind speed ratio $\alpha' = \alpha_{event} - \alpha_{clim}$, we will instead calculate the relative difference from climatology as $(\alpha_{event} - \alpha_{clim})/\alpha_{clim}$. This will make the effect size much more clear in the discussion.

L364: What is exactly meant with "streamline"?
Streamlines are curves that are everywhere tangent to the local velocity in a fluid flow. The wind streamline intersecting Point Barrow traces out the path from which the winds intersecting Point Barrow originate. We will add a sentence in the revised manuscript to clarify this: "Wind streamlines (curves tangent to the local wind velocity) are displayed in Fig. 9a to trace the direction and extent of wind forcing across the region. North of Alaska, the wind streamline intersecting Point Barrow marks the transition…" We point this out in the text and figures because this streamline delineates the portions of the wind circulation that force the ice against the Alaskan coast to the east of Point Barrow and along the coast west of Point Barrow.

L435: To me it was not really clear what is meant with "a synoptic center aligns with a known center of action".
We will restructure the paragraph preceding this sentence in an effort to provide more clarity, and also provide examples of what is meant within the sentence: "Even as the center of a synoptic forcing system (e.g. a passing high) aligns with a known center of action (e.g. the mean Beaufort high position), regional wind differences between the two aligned forcing systems can yield pronounced differences in the large-scale ice circulation."

L 439: "O (500 km)" also in L 481.
L 439: We will replace "O(500 km)" with "on the order of 500 km"
L 481: We will replace "O(500 km) offshore" with "approximately 500 km from shore"

The Discussion (4) is very extensive and can be shortened, I think. Some arguments seem to repeat.
We thank Dr. Willmes for this feedback, and will shorten the discussion sections to improve the flow and clarity of the manuscript.

LL 522-528. The description in this paragraph was not clear to me.
We have modified this paragraph to clarify that we were emphasizing the frequency of these events in winter (occurring about 20% of the time) as a motivation for why these seemingly transient events need to be represented accurately in dynamic ice models. The paragraph will read: "We identified 82 distinct event sequences, nearly one six-day sequence per month in the analysis period. Cumulatively, these events span approximately one-fifth of winter periods (January-April) between 2000 and 2020. This is a conservative estimate, as nearly 40% of the total 135 identified lead opening events overlapped with the distinct

sequences included in the ensemble. Given the frequency of these episodic events throughout the consolidated season, their associated ice drift patterns must be represented accurately in models in order to support predictions of ice transport on seasonal timescales."

Section 5: Is also very extensive, could maybe be shortened.
We appreciate this suggestion and will shorten the summary and conclusion section in the revised manuscript.

**Technical corrections**
None.

---

## Author Comment (AC2)

**Response to Referee Comments**

We would like to thank Referee #2 for their consideration of our manuscript and for providing constructive comments and suggestions. With their feedback the presentation of the manuscript will greatly improve. In this document we will address the referee's feedback point by point. Below we show referee comments in orange text and provide our response in light blue text. In dark blue text, we list the changes in the manuscript that address these comments.
* * *
**Referee #2 Comment on tc-2023-9**

Referee comment on "Atmospheric highs drive asymmetric sea ice drift during lead opening from Point Barrow" by MacKenzie E. Jewell et al., The Cryosphere Discuss., https://doi.org/10.5194/tc-2023-9-RC2, 2023

This paper analyses the atmospheric conditions during observed sea-ice lead openings from Point Barrow from 2000 to 2020. The authors use the ERA5 reanalysis to generate an atmospheric composite describing the mean atmospheric state during a lead opening. This is augmented by the observed ice drift from the Polar Pathfinder sea-ice motion product. The authors analyse the mean atmospheric and sea-ice state during a lead opening, concluding that such events are primarily driven by strong winds associated with an anti-cyclone over the Beaufort Sea, driving the ice along the Canadian and Alaskan coast, causing a lead to open at Point Barrow, which acts as a focal point for the stresses in the ice. They also analyse the ice response to the different wind directions observed when the lead opens up.

The paper is interesting, well-written, and well deserving of publication in The Cryosphere. The approach is novel, interesting, and well-suited to analyse the lead formation, both of Point Barrow and in general. The paper is very informative, and there is much information there. It is also well-written and readable. I only have one general and a few specific comments on the paper and recommend publication once those are addressed.

We thank the referee for their assessment of the manuscript. Below we will address the general comments and specific comments on the manuscript, and our plan to address these comments in the revised manuscript.

**General comment:**

There's an anomaly in SLP associated with lead openings (e.g. figure 4). But this doesn't seem very dynamically relevant. So the anomaly in the SLP gradient (e.g. figure 7) should be highlighted instead.

We thank the reviewer for pointing out this distinction: the magnitude of SLP is not the primary mechanism that controls the dynamics associated with lead opening. Since the surface winds are the primary mechanism through which the atmosphere dynamically forces the ice, we highlight the differences in wind direction associated with these events compared to the climatological wind distribution in Figure 4b. Changes in wind direction appear most dynamically relevant to these events and are therefore discussed throughout the text. Associated anomalies in SLP gradient are included in Figure 7 and described in the text as a causal mechanism that produces onshore wind anomalies in the ensemble

sequence. This is highlighted in the discussion where we describe how differences in the shape of the weather system (deviations in SLP gradient direction from the average Beaufort High) produce the average ensemble event. We avoided referencing the SLP gradient specifically in Figure 4 since this is equivalent to describing the winds they produce. As ongoing research for the lead author's doctoral work, we believe the gradient SLP relationship to sea ice forcing contains a phase lag in relationship to energy buildup relative to the coast - which will complicate the paper and its interpretation beyond the current scope. We therefore feel it important to keep this point out for now and consider it a matter for future work.

We have chosen to include the SLP distribution in Figure 4 (and show SLP fields in other figures) to highlight that high pressure atmospheric systems transiting the Beaufort and Chukchi Seas are the source of the wind patterns (i.e. SLP gradients) relevant to lead opening. In the revised manuscript, we shorten and clarify the discussion and conclusion sections of the paper to improve flow and clarity. We expect this will help to highlight the roles of high-pressure systems as the source of the activity, and winds acting through SLP gradients as the way in which the highs directly force the ice during these events.

**Specific comments:**
L52: New paragraph at "Landfast ice ..."L52: No need for brackets explaining landfast ice
We will remove the bracketed description in the revised manuscript.

L63: Change "translating" to "traversing" (for example).
We will change "translating" → "traversing"

L92: MODIS lead detection is impressive under ideal conditions. But I would have liked to learn more about its ability to detect leads under less-than-ideal conditions. There is no mention of cloud cover problems, for example.
Issues detecting leads under cloud cover were addressed later in the manuscript, but we thank the referee for reminding us that the limitations of MODIS in detecting leads (due to clouds) should be stated in this section as well. We will add a statement clarifying the limitations after line 92: "Cloud cover can increase the minimum resolvable lead width or even mask surface conditions altogether, preventing lead detection."

L235: Interesting that the winds strengthen after lead opening. It sounds like a selection bias, but how that would work is not immediately apparent. Should be addressed in the discussion.
In a previous publication (Jewell and Hutchings, 2023; https://doi.org/10.1029/2022GL101408) discussing the wind forcing associated with lead opening (from ERA5 reanalysis between 1993-2013), we have found that surface winds over the Beaufort Sea originating from the east are 40% faster than winds from the north, south, and west on average. Thus, the increase in wind speed following lead opening in the ensemble sequence here is likely associated with the tendency for winds to shift easterly (and consequently to strengthen) during and following opening as the high progresses

eastward following opening. Wind speed does not appear to be a key control on lead opening in these cases, as some events occur under very low wind speeds. This was similarly demonstrated by Lewis and Hutchings (2019, https://doi.org/10.1029/2018JC014898) who did not find a threshold for wind speed required to form lead opening patterns along the Alaska coast. We are hesitant to address the increase in wind speed in the discussion out of concern that it may imply there is a greater relative importance of wind speed during these events than there appears to be. We will address this in the results section where the ensemble is discussed by modifying line 235 as follows:  "In the days following lead opening, winds  shift westward over the Beaufort ice pack as the high-pressure system continues traveling eastward. The winds also strengthen as they rotate westward, as is common in the Beaufort Sea where westward winds blow stronger than winds blowing toward other directions (Jewell and Hutchings, 2023)."

L276: Mention that \alpha is essentially the Nansen number (if you ignore ocean currents). As it stands, its appearance here can seem a bit random.
We originally omitted this description so as not to imply to the reader that this ratio was calculated in the way that wind factor or Nansen number would be calculated, but agree that the purpose of its use here could be unclear as a result. We thank the referee for this suggestion and have added a sentence to clarify this point. "This ratio is similar to the wind factor or Nansen number. However, rather than describing the instantaneous relationships between wind and ice drift, we calculate the ratio of the average wind and ice speeds at each location."

L330/Paragraph: Figure 8(b) needs a better explanation. Why do you take the projection of the ensemble drift onto the climatological one? What does this show us? You say it reveals "one of the most striking features of the ensemble event sequence", but this is lost on me. I feel like I do not understand something important here.
To make this point more clear, we adjust the figure and descriptions of Figure 8(b) in the revised manuscript (now Figure 9b) as follows: "Underlain is the  component of the six-day average ice drift vector anomalies (u')  aligned along the climatological drift vectors (U), calculated as (u′ · U)|U|−1. This quantifies how much the ensemble ice drift  contributes to the climatological ice circulation over the six-day sequence. Positive  values show where the anomalies are aligned along  the typical flow direction, corresponding to a strengthening  of the  climatological drift pattern during the events. Negative values represent vector anomalies aligned against the climatological vectors and a weakening of the climatological drift."

We remove the phrase  "one of the most striking features of the ensemble event sequence" and add more specific language in the following paragraphs to describe what Figure 9(b) is depicting: "As a result of these varying wind-driven ice-coast interactions, the ensemble lead opening sequence's contribution to the climatological Beaufort Gyre circulation varies regionally. Figure 9(b) demonstrates this, showing a pronounced zonal asymmetry in Gyre

strength along the Alaskan coast during the ensemble sequences. The western flow of the Beaufort Gyre is strengthened by 1-2 cm/s on average throughout the ensemble, strengthening ice advection across the Chukchi, East Siberian, and Laptev Seas. Strengthening is greatest along the Chukchi coast of Alaska, where the climatological drift is enhanced by over 2 cm/s. The change in coastline orientation at Point Barrow marks the transition between where these events strengthen and weaken the climatological patterns of winter ice transport. Thus, in contrast to the strengthened Gyre circulation west of Point Barrow, in the Beaufort Sea the Gyre is weakened by up to 1 cm/s during the ensemble sequence."

L344: I found this to be the most interesting section! Seeing how the leads open along the wind streamline was particularly interesting. This indicates that the wind direction and topography combination is the controlling factor in lead formation off Point Barrow and that ice strength should have a relatively small impact. This also indicates that modelling such events should be pretty straightforward, but this is not the case. It also contradicts the results of Rheinlænder et al. (2022), who found that thinner ice broke up more easily. So there's food for thought here, which is highly appreciated.

Presentation-wise, I would have liked to see the mean wind field rather than the isolines you show. It's confusing that the in the easterly case, the lead opens perpendicular to the isobars, but this is actually along the wind streamline. I would also note that a more significant role of ice dynamics is indicated where the lead deviates from the wind streamline. Finally, I would remove the westerly case. You say you include it for completeness, but with such few members of your ensemble, I think it's safer to leave it out.

We thank the referee for this constructive feedback. We were also very interested to see the ways in which these observations align with and also contradict recent modeling studies. In Figure 9, we included pressure fields to show how offsets in high-pressure patterns can produce the wind fields that drive these events. We agree that showing the wind fields would be useful, so will include wind streamlines as an overlay on the bottom row of the figure, as opposed to the ice drift vector anomalies which are not key to the story. We also appreciate the suggestion to remove the westerly case, and will do so in the revised manuscript. We would be hesitant to state that ice dynamics play a more significant role when the lead orientation deviates from the local wind direction, since the leads may be forming under differing mechanisms when opening at different orientations relative to the winds (e.g. shear vs tension), each with a significant role of the internal ice dynamics. The differences in internal ice stress states in the vicinity of the lead openings are a very interesting point that were a motivating factor for developing this study. However, as we found the observational data employed were not sufficient to tease out specific stress states with certainty, we will leave specific statements regarding the momentum balance in the vicinity of the leads (and consequently the modes of failure that produce the patterns) to future work where stress states may be more accurately estimated.

L421: This paragraph belongs in the introduction rather than here.

This paragraph will be moved into the introduction in the revised manuscript.

L420: This is a nice and interesting discussion. But I would start by looking at the mean state (the ensemble means) and then discuss that there are variations from those. That order makes more sense. Simply moving sections around a bit would do.
We appreciate the suggestion to restructure this discussion section. We will begin with the ensemble mean then move to the cross-event variability in the revised manuscript.

L470: "… and in summer …" - I guess the "and" should not be there.
With this sentence, we were aiming to state how the linear relationship between winds and ice drift is more accurate in summer and in the Central Arctic (away from coastlines) since internal ice stresses are lower. However, we see that the wording of the sentence made this unclear. We will rearrange this paragraph to clarify this point and change the sentence to state: "Linear relationships between winds and ice drift successfully describe ice motion in summer when ice concentrations and internal stresses are low. As internal stresses increase during consolidation of the ice pack in winter, the ratio of ice drift to wind speeds decreases and correlation between winds and ice drift weakens, especially near coastal boundaries (Thorndike and Colony, 1982)."